# Genomic and metabolic adaptations of biofilms to ecological windows of opportunity in glacier-fed streams

Susheel Bhanu Busi [1,5], Massimo Bourquin [2,5], Stilianos Fodelianakis[2,5], Grégoire Michoud [2], Tyler J. Kohler[2], Hannes Peter[2], Paraskevi Pramateftaki[2], Michail Styllas [2], Matteo Tolosano[2], Vincent De Staercke[2], Martina Schön [2], Laura de Nies[1], Ramona Marasco [3], Daniele Daffonchio[3], Leïla Ezzat[2], Paul Wilmes [1,4✉] & Tom J. Battin [2✉]

In glacier-fed streams, ecological windows of opportunity allow complex microbial biofilms to develop and transiently form the basis of the food web, thereby controlling key ecosystem processes. Using metagenome-assembled genomes, we unravel strategies that allow biofilms to seize this opportunity in an ecosystem otherwise characterized by harsh environmental conditions. We observe a diverse microbiome spanning the entire tree of life including a rich virome. Various co-existing energy acquisition pathways point to diverse niches and the exploitation of available resources, likely fostering the establishment of complex biofilms during windows of opportunity. The wide occurrence of rhodopsins, besides chlorophyll, highlights the role of solar energy capture in these biofilms while internal carbon and nutrient cycling between photoautotrophs and heterotrophs may help overcome constraints imposed by oligotrophy in these habitats. Mechanisms potentially protecting bacteria against low temperatures and high UV-radiation are also revealed and the selective pressure of this environment is further highlighted by a phylogenomic analysis differentiating important components of the glacier-fed stream microbiome from other ecosystems. Our findings reveal key genomic underpinnings of adaptive traits contributing to the success of complex biofilms to exploit environmental opportunities in glacier-fed streams, which are now rapidly changing owing to global warming.

[1] Systems Ecology Group, Luxembourg Centre for Systems Biomedicine, University of Luxembourg, Esch-sur-Alzette, Luxembourg. [2] River Ecosystems Laboratory, Center for Alpine and Polar Environmental Research (ALPOLE), Ecole Polytechnique Fédérale de Lausanne (EPFL), Lausanne, Switzerland. [3] Biological and Environmental Sciences and Engineering Division (BESE), King Abdullah University of Science and Technology (KAUST), Thuwal, Saudi Arabia. [4] Department of Life Sciences and Medicine, Faculty of Science, Technology and Medicine, University of Luxembourg, Esch-sur-Alzette, Luxembourg. [5] These authors contributed equally: Susheel Bhanu Busi, Massimo Bourquin, Stilianos Fodelianakis. ✉email: paul.wilmes@uni.lu; tom.battin@epfl.ch

Ecosystems and their constituent biota are finely tuned to the seasonal variations of their environment. This phenology is particularly pronounced in glacier-fed streams (hereafter GFSs), which are commonly enveloped by snow cover and darkness in winter, and subject to high flow and sediment mobilization in summer. Yet, ecological 'windows of opportunity' arise in spring and autumn[1,2] when nutrient (N, P) and light availability is elevated and streamflow is moderate[1–3]. During the onset of spring snowmelt, inorganic N that has accumulated from atmospheric deposition and concentrated at the snowpack surface is washed into GFSs[3,4], whereas hydrologic connectivity with various glacial sources (e.g., subglacial) can increase concentrations of phosphorus as the melt season progresses[4,5]. Following the height of the melt season in summer, discharge and turbidity decline in autumn, again elevating nutrient concentrations and light availability. These favorable conditions allow algae and cyanobacteria to rapidly develop into 'green oases' of phototrophic biofilms. Partially due to the absence of major terrestrial organic matter subsidies from the catchment, this punctuated exploitation of solar energy in an otherwise energy-limited ecosystem transiently forms the base of the GFS food web and ecosystem energetics[1,6]. Such windows of opportunity may therefore function as 'ecosystem control points'[7] with disproportionately high ecological processing rates affecting ecosystem dynamics relative to longer intervening time periods. These ecosystem control points are widely distributed across ecosystems and vary across spatial and temporal scales[7]. However, our understanding on the microbiology of the communities that facilitate ecosystem control points remains limited to date.

Owing to climate change, the mass balance and melting dynamics of mountain glaciers are rapidly changing worldwide, altering the annual distribution of runoff in GFSs[8]. Invigorated glacial melt increases discharge and sediment delivery, but after glaciers shrink past a certain point (i.e., 'peak water'), GFSs are likely to become warmer, less turbid, and less hydrologically dynamic[4]. These changes are almost certain to have substantial impacts on GFS ecosystem structure and function by either contracting or extending the duration of these windows of opportunity. It is therefore critical to understand how benthic biofilms operate during these times in order to predict how these ecosystems are likely to change in the future[4].

In streams, biofilms closely interact with the sedimentary environment[9]. For example, extracellular polymeric substances (EPS) produced by biofilms bind fine sediment grains together, which can locally stabilize substrata, reducing scour and vertical permeability[10]. Similarly, boulders resist flow-induced disturbance to promote biofilm growth[11], and if protruding through the water column, may also increase light availability to further facilitate photosynthesis. Therefore, it seems advantageous for phototrophic biofilms to colonize boulders, which can be regarded as islands of stability in otherwise highly unstable GFS channels. These islands may allow biofilm growth to locally persist beyond the typical windows of opportunity (at least until snow cover), drive ecosystem energetics (i.e., gross primary production)[12], and to sustain the GFS food web and related benthic biodiversity[4,13].

The relationships between phototrophs (such as algae and cyanobacteria), prokaryotes and fungi regulate nutrient and carbon cycling, and therefore represent a fundamental ecological interface in aquatic ecosystems. This interface (i.e., the phycosphere) has received substantial attention in pelagic ecosystems over the last decades[14–17], but less so in stream ecosystems. While early work on phototrophic biofilms colonizing the benthic zone in streams has highlighted the role of algal–bacterial interactions for carbon and nutrient fluxes[18,19], we do not currently understand the fine-scale mechanisms of such interactions. For example, cyanobacteria produce pigments that protect the biofilm as a whole against harmful UV-radiation[20], while mucilage-rich algal colonies (e.g., *Hydrurus* spp.) provide labile organic matter to heterotrophic microorganisms and facilitate their attachment. Such interactions may foster facultative interactions between photoautotrophs and other microorganisms, which, similarly to the phycosphere, may be particularly beneficial to microbial life in oligotrophic and harsh ecosystems such as GFSs. Unraveling the genomic and metabolic underpinnings of algal–bacterial relationships in biofilms helps to better understand the success of the biofilm mode of life in an extreme ecosystem.

Here we dissect the microbiome of GFSs and describe the genomic underpinnings of the adaptive mechanisms that potentially contribute to the success of complex biofilms. Using 16S rRNA and 18S rRNA gene amplicon sequencing, we assess the microbiome structure of biofilms associated with two sedimentary habitats that are common in GFSs, namely sandy sediments (i.e., epipsammic biofilms) and boulders (i.e., epilithic biofilms). We sampled geographically distant streams, transcending hemispheres (Southern Alps in New Zealand, NZ, and the Caucasus, CC), to draw more generalisable conclusions about microbiome structure and assembly. Furthermore, using genome-resolved metagenomics, we screen twenty-one epilithic biofilm microbiomes for energy pathways and cross-domain metabolic interactions. Our findings suggest the diversification of energy-acquiring pathways and metabolic interactions are relevant for epilithic biofilms to thrive during the ecological windows of opportunity, and beyond, within low-disturbance patches in GFSs. Moreover, our findings shed light on what the future biofilm mode of life in GFSs may look like as glaciers shrink and GFS ecosystems are predicted to become more autotrophic[4].

## Results and discussion

**Sedimentary habitats affect microbiome structure and assembly.** We used 16S rRNA and 18S rRNA gene amplicon sequencing to compare the microbiome structure of 48 epipsammic and epilithic biofilm samples from GFSs in NZ and CC collected during spring and autumn, respectively (Methods) (Fig. 1a; Supplementary Fig. 1a, b). These seasons broadly align with the windows of opportunity in these GFSs; however, we recognize that epilithic biofilms, in particular, may extend beyond these windows well into summer or until snow coverage. We found that both prokaryotic and eukaryotic communities differed between the two habitat types in terms of community structure and alpha diversity (Fig. 1b, c). Overall, taxonomic differences were even apparent at the phylum level, despite high inter-sample variability within the categories (Supplementary Fig. 1c, d). Geography (i.e., NZ versus CC) explained 11.5% and 12.9% of the variability in the prokaryotic and eukaryotic datasets (db-RDA, $p < 0.05$ for both datasets), while sedimentary habitats explained an additional 10% and 8.3% of the variability (db-RDA, $p < 0.05$ for prokaryotes and eukaryotes).

The estimated α-diversity (i.e., richness of amplicon sequence variants; ASVs) was higher for both prokaryotes and eukaryotes in epipsammic biofilms when compared to epilithic biofilms (2–3 fold differences, non-parametric $t$-tests, $p < 0.001$) (Fig. 1d, e). These observations are in accordance with findings by Tolotti and colleagues[21] where α-diversity of the epipsammic habitats were higher than the epilithic biofilms in rock glacier- and groundwater/precipitation-fed waters[21]. It is plausible that continuous dispersal and mixing facilitated by the transport of fine sediments from various upstream sources (e.g., the subglacial environment and adjacent soils) leads to the greater diversity of the epipsammic biofilms. Overall, our results unravel distinct microbiome structures for both sediment habitats within the

**Fig. 1 Sedimentary habitats affect microbiome structure and assembly. a** Representative images of sample collection indicating GFS and adjacent epilithic biofilm (left) with images of epilithic biofilms (right). Photo credits: Martina Schön and Matteo Tolosano. Ordination analyses of the epipsammic ($n = 27$ biologically independent samples) and epilithic ($n = 21$ biologically independent samples) biofilm based on prokaryote (**b**) and eukaryote (**c**) metabarcoding profiles from Southern Alps and Caucasus. Microbial richness across geographic locations and sample types in (**d**) prokaryotes and (**e**) eukaryotes. The statistical analyses was performed on 27 epipsammic and 21 epilithic samples using a two-sided non-parametric $t$ test. Bonferroni-corrected $p$ values are indicated by *, i.e., *** represents $p < 0.001$. Boxplots represent the median richness with the 25th and 75th quartiles observed within the samples.

same GFS reaches. This agrees with previous studies[21], and more generally with the relationship between streamed physical variation and spatial biodiversity dynamics[22,23]. Streambeds, including their biofilms, are understood as landscapes where dispersal among patches can shape biodiversity and resilience[24–26]. Therefore, we hypothesized that epilithic communities are partially structured by dispersal from epipsammic communities that typically dominate the GFS streambeds by area. Using Sloan's neutral community model[27], we instead found that the composition of the epilithic biofilms is not dictated by a source-sink relationship with the epipsammic communities (Supplementary Note). In other words, the epilithic biofilm communities are not determined by epipsammic communities that typically surround the boulders within the complex landscape of the GFS streambed.

**Metagenomics unveils the complexity of epilithic biofilms**. To unveil the full complexity of the epilithic biofilms, we performed whole genome shotgun metagenomics on 21 epilithic samples from four GFSs each in NZ and CC (Supplementary Fig. 1a, b); low biomass associated with sandy sediments precluded epipsammic biofilms from metagenomic analysis. Metagenomic sequencing, after quality filtering, yielded on average $1.2 \times 10^8$ ($\pm 1.4 \times 10^7$ s.d.) reads per sample which were assembled to obtain an average of $8.7 \times 10^5$ contigs per sample that were subsequently binned. Bacteria and eukaryotes dominated the biofilm communities across all samples (Supplementary Fig. 2a). Seventy-three (70 bacteria and three archaea) medium-to-high quality (>70% completion, < 5% contamination) metagenome-assembled genomes (MAGs) from a total of 662 MAGs formed the pool of the prokaryotes. As seen from the phylogenomic analysis, the high-quality MAGs ($n = 49$, >90% completion and <5% contamination) span the bacterial tree of life. Based on the phylogenomic analyses along with the taxonomic information (Fig. 2), we sought to further characterize these MAGs that could represent novel species or species that have not previously been reported (Fig. 2a). We found that only 30% of these high-quality MAGS were annotated up to the family level, whereas the remaining MAGs could be taxonomically labelled at the genus level. Only high-quality MAGs were used for the phylogenetic analyses to mitigate disparities arising from incomplete MAGs. Aggregated at the genus level, *Polaromonas* was both abundant and prevalent in the biofilms along with representatives of *Flavobacterium*, *Cyanobacteria*, and unclassified MAGs from the Bacteroidota and Candidate Phyla Radiation (CPR; *Patescibacteria*) (Fig. 2b). These taxa were found in over half of the samples, irrespective of geographic origin. The CPR bacteria have only recently been identified based on genomic data[28], and *Patescibacteria* specifically have been reported from oligotrophic ecosystems, including groundwater[29] and thermokarst lakes[30]. Their apparently minimal biosynthetic and metabolic pathways may help them dwell in these ecosystems, which is of equal relevance in GFSs.

Alongside these bacteria, archaea contributed less than 1% to the microbiome of epilithic biofilms, with representatives of Asgardarchaeota, Crenarchaeota and Nanoarchaeota. Intriguingly, the recently discovered lineages of Asgardarchaeota[31,32] have been reported from freshwater sediments, yet not from cryospheric environments. Algae, mostly diatoms and *Hydrurus* (Ochrophyta phylum), as well as dinoflagellata, were the most important photoautotrophs of the eukaryotic domain (Fig. 2c). The prevalence of *Hydrurus* (~87% relative abundance) underscores the function of these filamentous algae as a resource to higher trophic levels in GFS[33]. Our metagenomic insights further support the notion that phototrophic biofilms are highly diverse with representatives from all three domains of life[28].

In addition to the archaeal, bacterial and eukaryotic community members, we also found a diverse viral community associated with epilithic biofilms (Supplementary Fig. 2b). Most of the viruses were bacteriophages targeting abundant MAGs such as *Flavobacterium*, *Pseudomonas*, and *Bacillus* genera, but we also identified eukaryotic phages (i.e., *Paramecium bursaria* Chlorella virus). Few have studied viruses in stream biofilms to date[34], potentially because it was common wisdom that the biofilm mode of life protects bacteria from viral infection. While viruses have previously been shown to be abundant in glaciers[35,36], our findings provide evidence for a diverse and likely active viral community in GFS biofilms where they may influence bacterial growth and both carbon and nutrient cycling as on the glacier surface[35].

**Epilithic biofilms form the basis for a 'green' food web in glacier-fed streams**. Cyanobacteria and eukaryotic algae dominated the photoautotrophs in the epilithic biofilms and hence form the basis of the 'green' food web during the windows of opportunity. While these photoautotrophs are well known to use chlorophyll to capture solar energy, little is known on retinal-based phototrophy using rhodopsins in GFSs. Intriguingly, we found that MAGs from sixteen out of twenty phyla in the epilithic biofilms, including the abundant groups, such as Proteobacteria (*Polaromonas*) and Bacteroidota (*Flavobacterium*), encoded for (bacterio-)rhodopsins (Fig. 3a). These also included genes encoding for light-harvesting complex 1 (LH1), reaction centre (RC) subunits (*pufBALM*), and transcriptional regulators (*ppsR*) required for aerobic anoxygenic phototrophs along with rhodopsins as a signature of energy-limitation adaptations (Fig. 3a). Recently, rhodopsins were also reported to serve as a photoprotectant in *Flavobacterium* from glaciers[37]. Collectively, our findings unveil multiple strategies of photoautotrophy, which may help cyanobacteria and algae to maximize their utilization of solar energy and to thrive on boulders in GFSs.

In order to exploit the favorable habitat provided by boulders during and beyond the windows of opportunity in GFS, rapid growth may be advantageous for primary producers such as cyanobacteria. Moreover, functional independence from other microorganisms could allow them to seize environmental opportunities. To test this hypothesis, we assessed the relationship between projected times of growth (doubling time in hours) with the median KEGG pathway completion within each MAG. Given the partial completeness of the MAGs, including possibly missing metabolic modules, we performed a linear regression between median KEGG pathway completion and projected time of growth, accounting for MAG completion as a fixed effect. Strikingly, 86% of the cyanobacterial MAGs ($n = 38$ out of 44) exhibited decreased projected times of growth with an increase in median KEGG module completion per MAG ($r_s = -0.47$, Two way ANOVA, adj. $p < 0.05$). These observations suggest that when encoding all genes to form a complete KEGG pathway, phototrophic taxa within these epilithic biofilms may indeed grow rapidly and be self-sufficient, putatively autonomously from other microorganisms from other (micro)organisms and fostering growth.

Given the energetic constraints in GFSs, it would be beneficial for bacterial heterotrophs to interact with these photoautotrophic (micro)organisms for meeting their energy and nutrient demands. To investigate such cross-domain relationships, we used network analyses and identified key interacting taxa based on positively co-occurring nodes using all prokaryotic and eukaryotic MAGs (see Methods). Based on a null model assessment (see Methods), our interaction networks showed preferential attachment within the nodes, along with increased

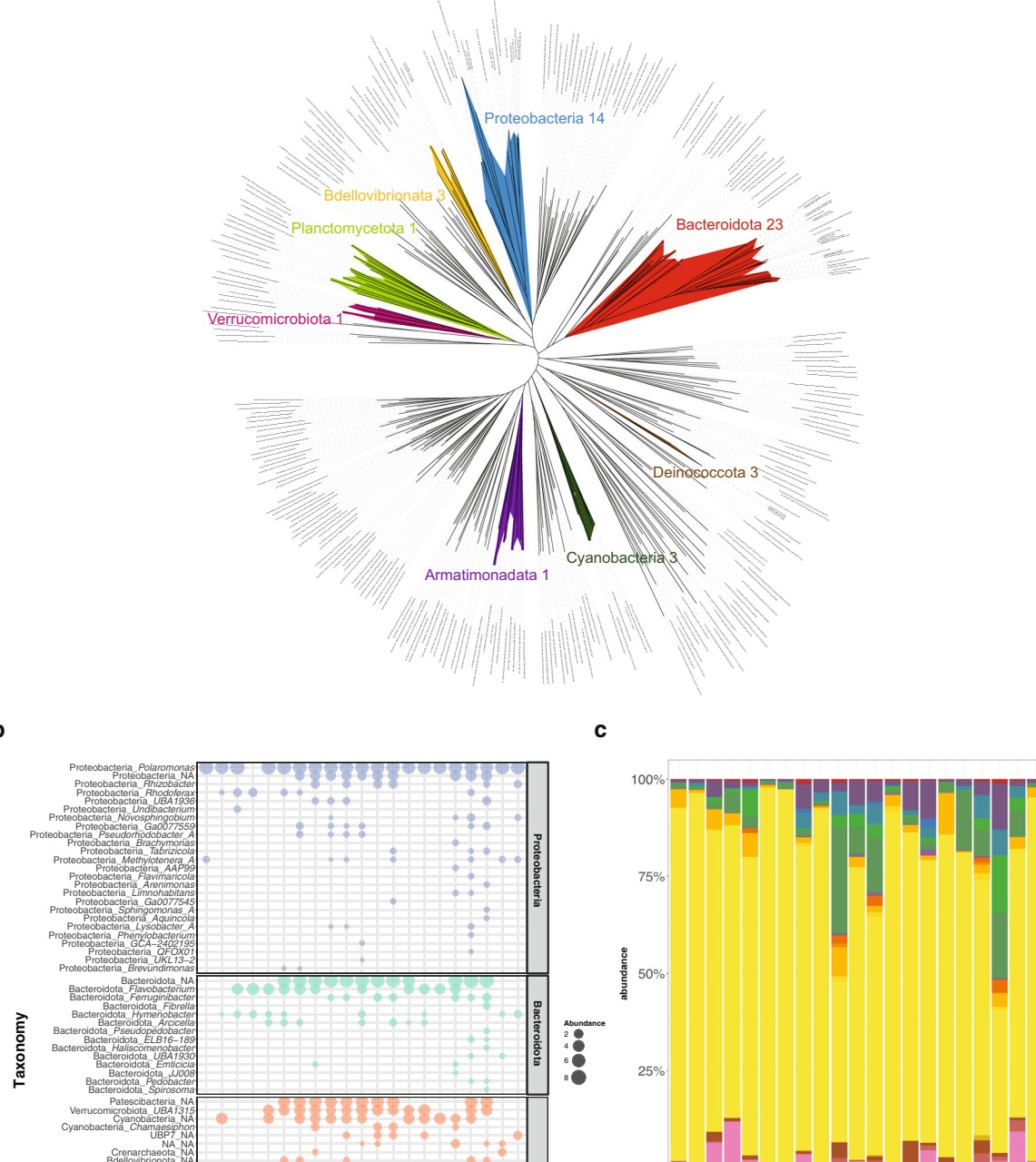

**Fig. 2 Metagenomics unveils the complexity of epilithic biofilms. a** Bacterial phylogenetic tree constructed using high-quality ($n = 49$, >90% completion and <2% contamination) MAGs reconstructed from the epilithic biofilms. The numbers beside the phylum names indicate the number of high-quality MAGs assigned to the respective phylum. Only high-quality MAGs were used to mitigate phylogenetic disparities from incomplete MAGs. **b** Normalized abundance of reconstructed prokaryotic genomes, i.e., MAGs, from the epilithic biofilms. Taxonomy at phylum and genus levels is depicted. NA: unclassified genus. Samples from the Southern Alps are indicated in red, while those from Caucasus are shown in blue. Medium-to-high quality MAGs ($n = 73$) are depicted. **c** Eukaryotic relative abundance profile obtained from metagenomic sequencing across all epilithic biofilms samples.

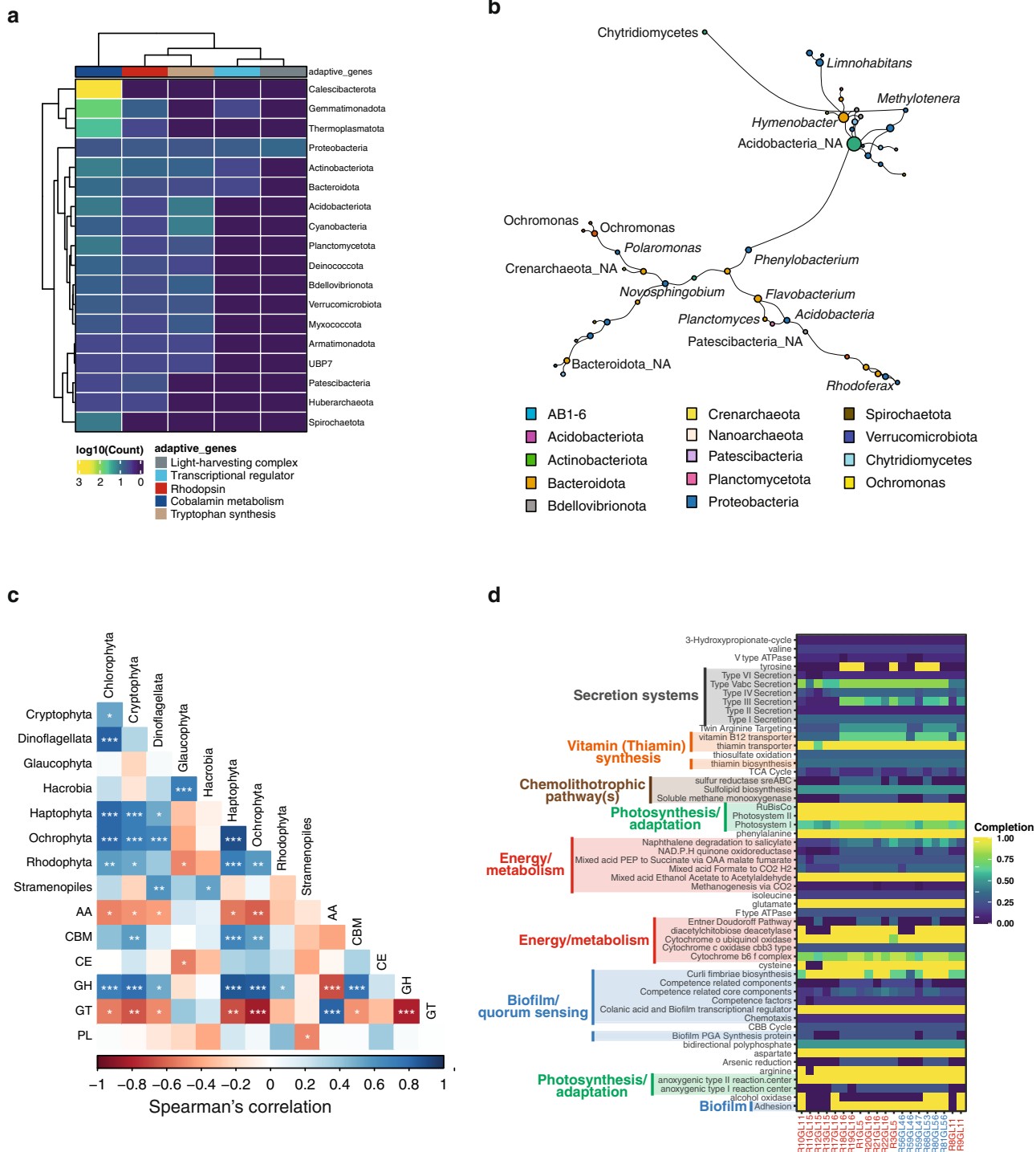

**Fig. 3 Epilithic biofilms are the basis for a 'green food chain'. a** Abundance of genes involved in energy production (light-harvesting complex, transcriptional regulator for phototrophy, and rhodopsin) and photo-heterotrophic interactions (cobalamin metabolism and tryptophan synthesis), across all prokaryotic phyla are represented in the heatmap. Values indicate the log$_{10}$ abundance per gene within the phyla. **b** Largest component of the co-occurrence network between pro- and eukaryotic MAGs. Each node corresponds to a MAG (pro- or eukaryote). Size of the node corresponds to degree centrality and the edges represent the positive coefficients of correlation between each node. Colour of each node represents the phylum annotation. NA: unclassified genus. **c** Spearman's correlation analyses of relative abundances of eukaryotic primary producers with the CAZyme abundances. CAZymes include AA auxilliary activities, CBM carbohydrate-binding module, CE carbohydrate esterases, GH glycoside hydrolases, GT glycosyltransferases, PL polysaccharide lyases. FDR-adjusted *p* values were estimated using the 'cor.mtest' function from the *corrplot* R package and are indicated by *, i.e., *<0.05, **<0.01, ***<0.001. **d** KEGG orthology (KO) pathways enriched in epilithic biofilms compared to publicly available cryospheric metagenomes were further assessed via KEGGDecoder for pathway completion and are displayed. The completeness of the pathways is indicated in the heatmap, per sample.

centralities (i.e., degree and edge-betweenness, Supplementary Fig. 3a, b), suggesting that the interactions within these networks were not random. More importantly, the largest connected component (based on degree and betweenness centralities) of the interaction network contained taxa spanning archaea, bacteria and eukaryotic domains (Fig. 3b and Supplementary Fig. 3b). Though *Acidobacteria* had a high degree of centrality, both *Polaromonas* and *Methylotenera* demonstrated strong interactions (>0.6 betweenness centrality) with primary producers (including eukaryotic algae) and fungi. Specifically, *Polaromonas* had a strong interaction with algae, while *Methylotenera* co-occurred with *Chytridiomycetes* (Fig. 3b). Interestingly, we found similarly connected nodes demonstrating cross-domain interactions within the largest component of the individual regions, i.e., NZ (Supplementary Fig. 3c, d) and CC (Supplementary Fig. 3e, f), albeit the two regions had varying numbers of edges (NZ = 205 and CC = 30). This suggests that inherent interactions within these GFS epilithic biofilms are conserved irrespective of geographic origins. These results also support our hypothesis of heterotrophic bacteria co-occurring with eukaryotes, primarily algae, for metabolic cross-feeding, similar to those occurring in the phycosphere[15].

Furthermore, our results hint at the existence of a more cryptic interaction in epilithic biofilms between the parasitic fungi *Chytridiomycetes* and algae (mostly *Ochrophyta*). Fungal parasitism on pelagic algae has been recently reported to be more important than expected, even with consequences for carbon and nutrient cycling as mediated by the fungal shunt[38,39]. The possibility of fungal parasitism on algae in epilithic biofilms further supports the notion of photoautotrophs forming the foundation of a complex food web in GFS ecosystems.

**Genomic underpinnings of algae–bacteria metabolic interactions.** As photoautotrophs grow and senesce, they increasingly exude intracellular material into their ambient environment, where it can be metabolized by heterotrophic bacteria through extracellular enzymes[40]. To explore this metabolic cross-feeding between bacterial heterotrophs and algae, we assessed the MAGs for genes encoding five common extracellular enzymes required for cleaving complex polysaccharides, phosphomonoesters and proteins[41]. Not unexpectedly, these genes were associated with bacterial heterotrophs rather than with the photoautotrophs (Supplementary Fig. 4), which suggests adapted genomic traits to meet specific metabolic needs of the heterotrophs. However, based on the presence of extracellular enzyme genes among Cyanobacteria, we cannot discount the possibility of mixotrophy in the epilithic biofilms (Supplementary Fig. 4b). Additionally, genes associated with mixotrophy, such as those encoding for auto- and heterotrophic pathways, were also found in other abundant members of the epilithic microbiome (e.g., Proteobacteria). The widespread occurrence of mixotrophy in planktonic communities[42], including members of the Cyanobacteria, and the ensuing food web dichotomy is considered as an adaptive strategy to oligotrophic and cold ecosystems (e.g., the polar sea[42] and alpine lakes[43]). Therefore, we argue that mixotrophy may also be an important trait of Cyanobacteria within GFS biofilms.

Carbohydrate-active enzymes (CAZymes) are the primary tools used by heterotrophic bacteria to initiate the degradation of polysaccharides, largely algae-derived in the GFS epilithic biofilms. To shed light on this potential trophic interaction identified through specific extracellular enzyme activities (EEAs), we tested if all the CAZymes in the metagenomes covaried with the abundance of eukaryotes. Overall, we found positive correlations between eukaryote abundances and CAZymes, particularly carbohydrate-binding modules (CBM) and glycoside

hydrolases (GH) (Supplementary Fig. 4d). More specifically, these correlations were particularly pronounced for GH and some of the algal groups (e.g., Ochrophyta, Haptophyta, Cryptophyta) that we found at relatively high abundances in the epilithic biofilms (Fig. 3c and Supplementary Fig. 4d). As some of these algae are known to copiously produce sulfated carbohydrates[44], we suggest a similar involvement of CAZymes (Supplementary Data 1) in relation to polysaccharide degradation in GFS epilithic biofilms as recently reported from Verrucomicrobia isolates[45]. Given that sulfated carbohydrates are more resistant to bacterial degradation than other carbohydrates[45], our findings suggest that they are still relevant to carbon turnover in an ecosystem that is inherently carbon limited.

In order to understand whether functions potentially geared towards cross-domain interactions were enriched in epilithic biofilms in GFSs, we compared the KEGG orthology (KO) annotations from our metagenomes to 105 metagenomes from a wide range of ecosystems (Supplementary Data 2). Strikingly, we found that whole metagenome comparisons revealed that KOs associated with quorum sensing, vitamin B12 (cobalamin) transporters and thiamine biosynthesis were enriched in epilithic GFS biofilms compared to other ecosystems (Supplementary Data 3). The associated pathways and their completion levels were evaluated using KEGGDecoder (Fig. 3d; Supplementary Fig. 5) indicating a high completion of pathways associated with cross-domain interactions. These findings are in line with previous genomic insights into algal–bacterial interactions[46,47], specifically with the observed upregulation of vitamin biosynthesis in bacteria (*Halomonas*) growing in the presence of algal extracts.

Furthermore, several MAGs were found to encode genes (e.g., quorum sensing, cobalamin metabolism, tryptophan synthesis) potentially facilitating algal–bacterial interactions (Fig. 3a). Particularly, cobalamin metabolism may be relevant for nutrient acquisition in algal–bacterial relationships[48], whereas tryptophan was reported as a key signalling molecule involved in interactions between bacteria and associated phytoplankton[16,49]. Collectively these genomic insights stress cross-domain interactions as an adaptive potential that the epilithic microorganisms have developed to exploit the window of opportunity in GFSs.

**Energy acquisition and biogeochemical pathways in epilithic biofilm MAGs.** The dominance (~88%) of MAGs encoding for organic carbon metabolism suggests a 'baseline' heterotrophy in GFSs likely supported by organic carbon subsidies from melting glaciers[6,50,51] 'green food web' during the windows of opportunity, potentially sustaining metabolic interactions between primary producers and heterotrophs. Given the notoriously low concentrations of dissolved organic carbon in GFSs[50,52,53], including our study sites in NZ ($96.18 \pm 21.35\ \mu g\,C\,L^{-1}$) and CC ($221.36 \pm 31.01\ \mu g\,C\,L^{-1}$), we suggest that the 'green food web' dominates over allochthonous subsidies.

Exploring the gene repertoire of the epilithic biofilms, we found that Cyanobacteria were one of the largest bacterial contributors to carbon fixation along with Bacteroidota and few *Gammaproteobacteria* (Fig. 4a). An in-depth analysis across the 662 MAGs revealed that 583 MAGs encoded genes involved in organic carbon oxidation, while 120 MAGs encoded genes involved in $CO_2$ fixation. In line with the above findings, the majority of these MAGs was identified as Cyanobacteria along with few other phyla such as Proteobacteria, Asgardarchaeota, Crenarchaeota and Huberarchaeota. We also note that 351 MAGs encoded genes for fermentation (Fig. 4b) spanning several phyla, including Actinobacteriota, Bacteroidota, Patescibacteria, Planctomycetota and Verrucomicrobiota.

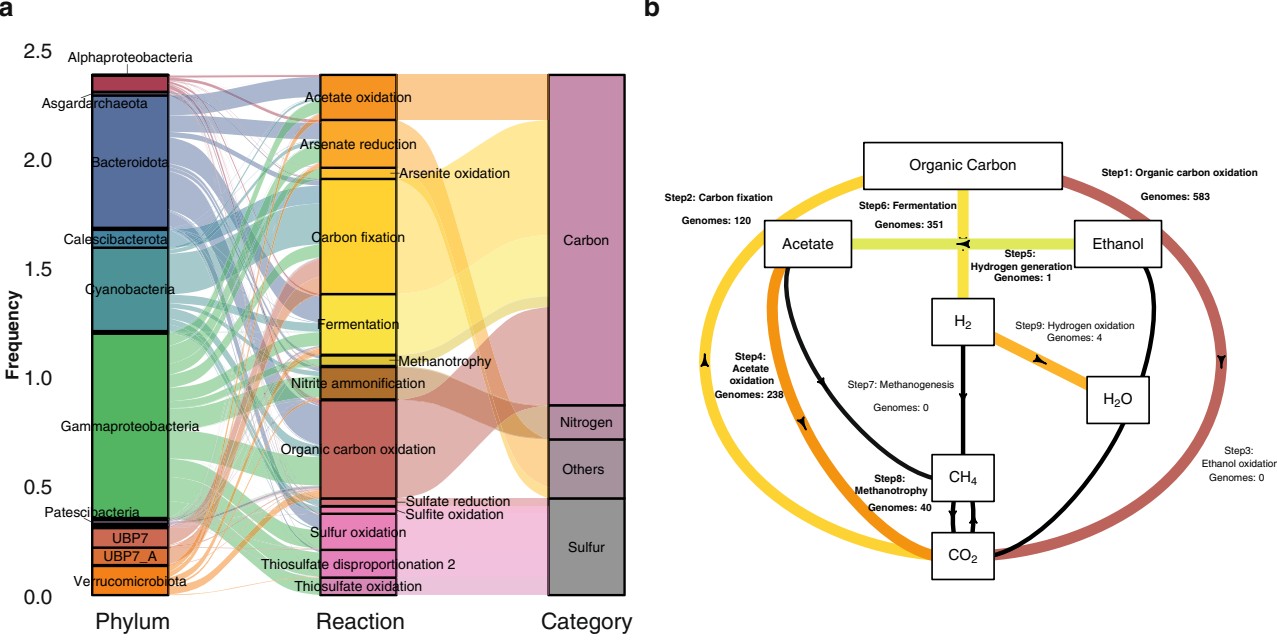

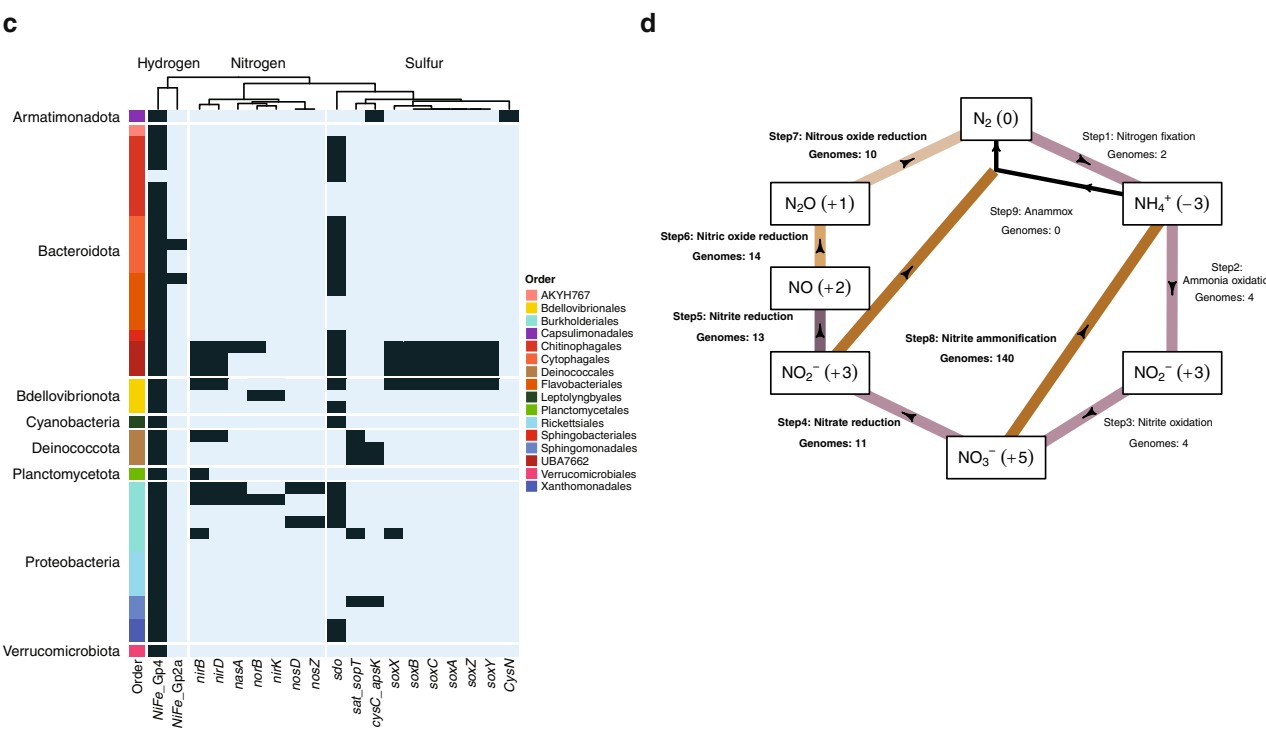

**Fig. 4 Functional redundancies across MAGs enable diverse energy acquisition and biogeochemical pathways. a** The alluvial plot represents the metabolic pathways identified within all prokaryotic MAGs, with the respective taxonomic classification and category of nutrients. **b** Total number of MAGs encoding genes for and involved in the Carbon cycle (Methods) are depicted in the flow gram created using a modified script from METABOLIC[109]. Each sub-pathway is indicated as a step with the corresponding number of genomes encoding the respective genes. **c** Phylum and order-level distributions of chemolithotrophic (hydrogen, nitrogen and sulfur) pathways with the respective gene copies per pathway are depicted in the heatmap. **d** Flow diagram indicating the MAGs encoding for pathways in the nitrogen cycle (Methods). Each sub-pathway is indicated as a step with the corresponding number of genomes encoding the respective genes.

For biofilms to thrive in GFSs, particularly during the windows of opportunity, it appears opportune to diversify the exploitation of energy sources. Therefore, we performed an in-depth characterisation of chemolithotrophic pathways to explore the potential role of minerals derived from the glacial comminution

of bedrock as an energy source for microorganisms[54]. The prevalence of the *sox* gene cluster in representatives of the Bacteriodota (UBA7662) and Bdellovibrionota reveals the potential importance of inorganic sulfur oxidation in epilithic biofilms. This notion is supported by the broad occurrence of

sulfur dioxygenases (SDOs) across the various phyla that facilitate sulfur oxidation (Fig. 4c). Interestingly, Tranter and Raiswell suggested that sulfates derived from sulfide oxidation in comminuted bedrock[55] potentially increase sulfur availability and acquisition in glacial meltwaters[56]. Sulfide oxidation can stimulate carbonate weathering with the resulting $CO_2$ potentially being fixed by algae and cyanobacteria in the epilithic biofilms—a link that appears relevant given that GFSs are often undersaturated in $CO_2$[57]. Furthermore, we found that almost all MAGs encoded for group IV hydrogen dehydrogenases (NiFe_Gp4; Fig. 4c), which potentially serve as an alternate energy acquisition pathway. Hydrogen dehydrogenases have recently been reported to support primary production in various glacial and other extreme environments[58,59]. This suggests that lithogenic hydrogen may also contribute energy to bacteria within the epilithic biofilms.

Genomic insights into the nitrogen cycle revealed the Dissimilatory Nitrate Reduction to Ammonium (DNRA, or nitrite ammonification) and, to a lesser extent, denitrification, as major pathways (Fig. 4d). Relatively little is known regarding these two competing pathways in stream biofilms or sediments[60], particularly in GFSs. This is in line with other ecosystems where DNRA is favoured over denitrification when alternate electron donors prevail over nitrate[61]. For instance, predicting metagenomes from 16S rRNA sequences, Ren et al.[62] found DNRA to be an important pathway in GFSs, suggesting that bacteria use inorganic nitrogen more as an energy source than a source for biosynthesis. Our analyses revealed Burkholderiales (Gammaproteobacteria) as the largest contributor to nitrate assimilation and ammonia-oxidation genes (Fig. 4a, c). DNRA, if not conducive to $N_2O$ production, would enhance nitrogen recycling within epilithic biofilms through ammonia assimilation by algae and cyanobacteria, for instance. Our genomic evidence for nitrogen recycling that potentially overwhelms nitrogen losses through denitrification is corroborated by flux measurements from microbial mats in Antarctic GFSs[63], and highlights recycling as a strategy to cope with nutrient limitation in glacier ecosystems[63–65].

Strikingly, we found only few MAGs, mostly belonging to Deinococcota, Gammaproteobacteria, Beijerinckiaceae and Crenarchaeota, involved in the oxidation of ammonia and nitrite, potentially leading to the accumulation of nitrate. The involvement of archaea would be in line with recent studies showing ammonia oxidation by archaea in Arctic soils[66] and with the observation that archaea couple ammonia oxidation with biomass formation (i.e., via $CO_2$ fixation)[67]. Our finding that archaeal MAGs encode for carbon fixation genes (Fig. 4b) further highlight their role in ammonia oxidation and biomass accrual in epilithic biofilms. Overall, the overlap of metabolic capacities within the MAGs suggests that the epilithic biofilms efficiently recycle carbon and nutrients. Internal recycling in stream biofilms is thought to be facilitated by increased residence times of water and contained solutes within the biofilms compared to the overlying water[68], which is certainly an advantage in a losing ecosystem such as GFSs.

**Genomic underpinnings of adaptation to the extreme GFS environment**. The GFS environment is extreme as illustrated by near-freezing temperatures, high UV-radiation, and high flow velocities. To assess potential adaptive traits of bacteria dwelling in epilithic biofilms, we first performed a phylogenomic analysis of Polaromonas spp., one of the most abundant and prevalent genera in the studied GFSs. Our analysis revealed that a few of the GFS Polaromonas formed clades that are distinct from Polaromonas identified in other environments (Methods), thus potentially comprising novel 'species' (Fig. 5a). This

phylogenomic pattern indicates that Polaromonas has evolved traits that facilitates its success in GFS, both in NZ and CC. To identify such traits, we created a pangenome and performed an enrichment analysis for clusters of orthologous genes. We found three categories that were significantly enriched in GFS Polaromonas compared to those from other environments (Supplementary Data 4). Two categories are related to defense mechanisms, both general and transcription, and one to energy production (Fig. 5b). It is plausible that these mechanisms are related to high UV-radiation[69,70] and oxidative stress[71], as well as to cold stress responses as previously reported from other bacteria[72–74]. Furthermore, the presence of CRISPR-Cas proteins in the enriched clusters of orthologous genes (COGs) hint at defense mechanisms against phages (Supplementary Data 4), which we showed to be present in the epilithic biofilms. This is in accordance with reports demonstrating that cryospheric bacteria (such as Janthinobacterium spp.) develop defense strategies, including biofilm formation[75] and extracellular vesicle formation[76] to escape viruses. On the other hand, the transcription of 'defense mechanism' genes have been linked to cold adaptation in psychrophiles[72]. Cold-shock proteins regulate transcription at low temperature, while genes involved in membrane biogenesis[77] and membrane transport proteins[78], several of which are also enriched in the GFS Polaromonas genomes, are up-regulated. For example, in the psychrophilic Colwellia psychrerythraea 34H, adaptation to cold includes the maintenance of the cell membrane in a liquid-crystalline state via the expression of genes involved in polyunsaturated fatty acid synthesis[79]. Similarly, ATP-driven or proton motive secondary transport systems have been associated with solute transfers across membranes in bacteria and archaea as an adaptation to the cold[78].

Our insights into the adaptive potential of Polaromonas to the GFS environment prompted us to expand our search for adaptive traits across all MAGs from the epilithic biofilms. Querying for 76 genetic traits spanning nine categories related to cold adaptation[73], we indeed found distinct patterns of genomic adaptation across MAGs (Fig. 5c). Several MAGs encoded for genes associated with membrane and peptidoglycan alterations, cold and heat shock proteins, oxidative stress, and transcription/ translation factors alongside DNA replication and repair. While all major phyla encoded for adaptive traits related to the outer membrane and cell wall, Proteobacteria were the predominant group with an overall higher copy number of genes (~5 copies/ genome), albeit insignificant compared to other phyla, involved in counteracting osmotic and oxidative stress. This was followed by Bacteroidota, Cyanobacteria and Actinobacteriota with three, two and two copies per genome respectively. Interestingly, we found that Patescibacteria MAGs had significantly lower copies of cold adaptation genes, whilst both Actinobacteria and Asgardarchaeota demonstrated a significantly higher number of osmotic stress genes (Supplementary Data 7). This is in line with metagenomic studies reporting an enrichment of sigma B genes in Antarctic mats, allowing for surviving severe osmotic stress during freezing[74]. Similarly, Psychrobacter arcticus[80] and Planococcus halocryophilus Or1[81] were shown to have specific genomic modifications, particularly with genes involved in putrescine and spermidine accumulation, both of which are associated with alleviating oxidative stress. Furthermore, MAGs from Proteobacteria were characterized by a high prevalence of genes potentially expressed in response to stressors, such as UV and reactive oxygen species (Fig. 5c).

Our genomic insights into possible adaptive traits of epilithic microorganisms may also contribute to our understanding of their adaptation beyond the windows of opportunity when the GFS environment is even harsher. In fact, with the onset of winter and during winter, GFSs partially freeze and become snow-

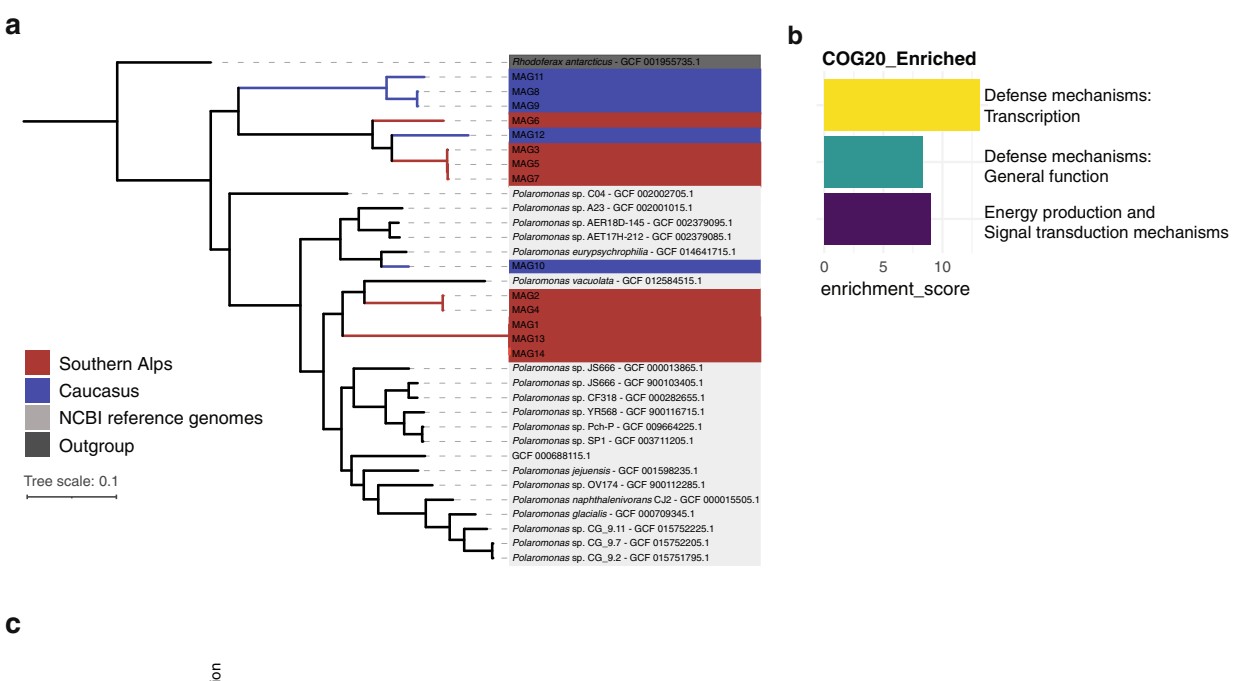

**Fig. 5 Genomic underpinnings of adaptation to the extreme GFS environment. a** Phylogenomic tree based on *Polaromonas* genomes recovered from Southern Alps (red) and Caucasus (blue) along with publicly available genomes (grey) and an outgroup (*Rhodoferax*, dark grey). **b** Clusters of orthologous (COG20) group pathways enriched in epilithic biofilms MAGs compared to the reference genomes are depicted in the barplot. **c** Heatmap representing the abundance of genes involved in cold adaptation. Taxonomy at phylum and order levels is depicted. Columns indicate clusters of orthologous groups associated with adaptive genes.

covered thereby inhibiting primary production. Mixotrophy as observed within the Cyanobacteria (Supplementary Fig. 4b) would be advantageous during these periods.

Furthermore, it is recognized that cell membrane alterations and lipid composition allow for withstanding cold conditions[73]. Our observations regarding several MAGs encoding genes associated with membrane and peptidoglycan alterations are concordant with previous reports of increased membrane fluidity in *Psychrobacter arcticus* 273–4[80], *Sphingopyxis alaskensis*[82], and *Pseudomonas extremaustralis*[83]. Simultaneously, at low temperatures oxygen solubility increases, potentially generating reactive oxygen species and subsequently leading to oxidative stress[84]. As reported above, we observed several MAGs encoding genes to counteract this phenomenon which may be even more critical as temperatures may decrease outside of the 'warmer' windows of opportunity. Overall, the diversity of the cold adaptation genes and their potential mechanisms within MAGs support the notion that these taxa are potentially equipped to deal with the even harsher GFS environment outside the windows of opportunity.

In conclusion, our genome-resolved metagenomic analyses have set the stage for a mechanistic understanding of how the diversification of energy and matter acquisition pathways, metabolic interactions, and genomic adaptations to harsh environmental conditions allow GFS biofilms to persist and thrive during windows of opportunity and beyond. We acknowledge that a metagenomic time series outside and throughout windows of opportunity would be required to substantiate some of our observations. Nevertheless, our findings shed light on boulders as important habitats that confer stability to biofilms even outside the typical windows of opportunity. GFSs count among the ecosystems that are most vulnerable to climate change. Therefore, our findings open a window into the future of how microbial life, with a strong photoautotrophic component, may look like in GFSs when the environmental conditions become more favorable for primary producers as glaciers shrink.

## Methods

**Sample collection.** We sampled a total of eight GFSs from the Southern Alps in New Zealand Southern Alps and the Caucasus in Russia in early- and mid-2019, respectively, for a total of 27 epipsammic samples taken from sandy sediments and 21 epilithic biofilm samples from boulders adjacent to the epipsammic samples (Supplementary Data 5). In order to have comparable samples, the collection was largely constrained to the vernal and autumnal windows of opportunity, respectively. Epipsammic samples were collected from each GFS by first identifying three patches within a reach of ~5–10 m. From each patch, epipsammic samples were taken from the <5 cm surface of the streambed with a flame-sterilized metal scoop and sieved to retain the 250 μm to 3.15 mm size fraction. While three epipsammic samples were taken from each stream, epilithic samples were taken opportunistically from up to three boulders per reach (Supplementary Data 5) due to their heterogeneity within and among the streams due to the unequal presence of boulders in each GFS. Epilithic biofilms were sampled using a sterilized metal spatula. All samples were immediately flash-frozen in liquid nitrogen in the field and transported and stored frozen pending DNA extraction. Streamwater turbidity, conductivity, temperature, and pH were measured in situ during the sampling (Supplementary Data 5). Samples for the determination of streamwater dissolved organic carbon and inorganic nutrient concentrations were filtered through pre-combusted (450 °C) glass microfiber filters (GF/F, Whatman), frozen, and analyzed in the laboratory. DOC concentration was measured with a TOC carbon analyzer (Sievers M9 TOC Analyser, GE). Phosphate, ammonium, nitrite and nitrate were measured with a continuous flow injection analyzer (Lachat QuikChem 8500, methods 10-115-01-1-M (PO$_4$), 10-107-04-1-B (NO$_3$/NO$_2$) and 10-107-06-3-D (NH$_3$)) (Supplementary Data 5).

**DNA extraction and purification.** A previously established protocol[85] was used to extract DNA from all samples. Briefly, 5 g of epipsammic and 0.05–0.1 g of epilithic biofilm were subjected to a phenol:chloroform-based extraction and purification method. The differential input volume for the DNA extractions were established to account for the differences in biomass between the epipsammic and epilithic biofilms. The samples were treated with a lysis buffer containing SDS along with 0.1 M Tris-HCl pH 7.5, 0.05 M EDTA pH 8, 1.25% SDS and RNase A (10 μl: 100 mg/ml). The samples were vortexed and incubated at 37 °C for 1 h. Proteinase K (100 μl; 20 mg/ml) was subsequently added and further incubated at 70 °C for

10 min. Samples were purified once with phenol/chloroform/isoamyl alcohol (ratio 25:24:1, pH 8) and the supernatant was subsequently extracted with a 24:1 ratio chloroform/isoamyl alcohol. Linear polyacrylamide (LPA) was used along with sodium acetate and ice-cold isopropanol for precipitating that DNA overnight at −20 °C. For epilithic biofilms, the entire protocol was adapted to a smaller scale due to the availability of higher DNA concentrations compared to sediment. The former was treated with 0.75 ml of lysis buffer (instead of 5 ml for sediment) and all subsequent volumes of reagents were adapted accordingly (see supplementary material). Furthermore, a mechanical lysis step of bead-beating was necessary along with a lysis buffer to facilitate DNA release from the more developed epilithic biofilms. Due to the higher DNA yields, the addition of LPA was omitted from the DNA precipitation step. DNA quantification was performed for all samples with the Qubit dsDNA HS kit (Invitrogen).

**Metabarcoding library preparation and sequencing.** The prokaryotic 16S rRNA gene metabarcoding library preparation was performed as described in Fodelianakis et al.[86], targeting the V3-V4 hypervariable region of the 16S rRNA gene with the 341 F (5′-CCTACGGGNGGCWGCAG-3′) and 785R (5′-GACTACHVGGG-TATCTAATCC-3′) primers and following Illumina guidelines for 16S metagenomic library preparation for the MiSeq system. The eukaryotic 18 S rRNA gene metabarcoding library preparation was performed likewise but using the TAR-euk454F (5′-CCAGCASCYGCGGTAATTCC-3′) and TAReukREV3 (5′- CTTTCG TTCTTGATYRA-3′) primers to target the 18 S rRNA gene V4 loop[87]. Samples were sequenced using a 300-bp paired-end protocol partly in the Genomic Technologies Facility of the University of Lausanne (27 epipsammic samples) and partly at the Biological Core Lab of the King Abdullah University of Science and Technology (21 epilithic samples).

**Metabarcoding analyses.** The 16S rRNA gene metabarcoding data were analysed using a combination of Trimmomatic[88] and QIIME2[89] as described in Fodelianakis et al.[86], with the exception that here the latest SILVA database[90] v138.1 was used for taxonomic classification of 16S rRNA and 18S rRNA gene amplicons. Non-bacterial ASVs including those affiliated to archaea, chloroplasts and mitochondria were discarded from the 16S rRNA amplicon dataset in all downstream analyses. ASVs observed only once were removed from both 16S rRNA and 18 S rRNA amplicon datasets. Diversity analyses were performed in R using the *vegan*[91] and *metacoder*[92] packages. For non-metric multidimensional scaling (nMDS) and distance-based redundancy (db-RDA) analyses data were $log(x + 1)$ transformed and the *capscale* and *ordiR2step* (backwards direction, 200 permutations) functions from *vegan* were used. To test for a source-sink hypothesis from epipsammic to epilithic, the Sloan's Neutral Community Model[27] was used based on the R implementation developed by Burns et al.[93].

**Whole-genome shotgun libraries and sequencing.** All epilithic biofilm DNA samples underwent random shotgun sequencing following library preparation using the NEBNext Ultra II FS library kit[94]. Briefly, 50 ng of DNA was used for constructing metagenomic libraries under 6 PCR amplification cycles, following enzymatic fragmentation of the input DNA for 12.5 min. The average insert size of the libraries was 450 bp. Qubit (Invitrogen) was used to quantify the libraries followed by quality assessment using the Bioanalyzer from Agilent. Sequencing was performed at the Functional Genomics Centre Zurich on a NovaSeq (Illumina) using a S4 flowcell.

**Metagenomic preprocessing, assembly, binning, and analyses.** For processing metagenomic sequence data, we used the Integrated Meta-omic Pipeline (IMP)[95] workflow to process paired forward and reverse reads using version 3.0 (commit# 9672c874; available at https://git-r3lab.uni.lu/IMP/imp3)[96]. IMP's workflow includes preprocessing, assembly, genome reconstructions and additional functional analysis of genes based on custom databases in a reproducible manner. Briefly, adapter trimming is followed by an iterative assembly using MEGAHIT v1.2.9[97]. Concurrently, MetaBAT2 v2.12.1[98] and MaxBin2 v2.2.7[99] are used for binning in addition to an in-house method, binny[100], for reconstructing metagenome-assembled genomes (MAGs). Binning was completed by selecting a non-redundant set of MAGs using DASTool[101] based on a score threshold of 0.7. The quality of the MAGs was assessed using CheckM v1.1.3[102], while taxonomy was assigned using the GTDB-toolkit v1.4.1[103].

For the downstream analyses including identification of viruses, VIBRANT v1.2.1[104] was used on the metagenomic assemblies. The output from this was used to identify the viral taxa using vConTACT2 v0.9.22[105]. Independently, the viral contigs were also validated using CheckV v0.7.0[106]. To estimate the overall abundances of eukaryotes along with prokaryotes including archaea, we used EUKulele v1.0.5[107] with both the MMETSP and the PhyloDB databases, run separately, to confirm the detected eukaryotic profiles. To understand the overall metabolic and functional potential of the metagenome and reconstructed MAGs we used MANTIS[108]. Additionally, we used METABOLIC v4.0[109], metabolisHMM v2.21[110], and Lithogenie from MagicLamp v1.0 (https://github.com/Arkadiy-Garber/MagicLamp) to identify metabolic and biogeochemical pathways relevant for determining nutritional phenotypes of all MAGs along with the 'anvi-estimate-metabolism' function from anvi'o[111]. This information was manually validated

based on the different tools to identify which MAGs encode for the respective pathways. Subsequently, to determine the growth rates of prokaryotes, we used codon usage statistics for detecting optimization of genes that are highly expressed, as an indicator of maximal growth rates with gRodon v1.0[112]. All the parameters, databases, and relevant code for the analyses described above are openly available at https://git-r3lab.uni.lu/susheel.busi/nomis_pipeline and included in the Code availability section.

**Eukaryote assembly and binning**. To obtain eukaryotic MAGs, an alternate, custom pipeline (https://github.com/Mass23/NOMIS_ENSEMBLE/tree/coassembly) was established for coassembling the twenty-one epilithic biofilm sequence data with subsequent binning. Individual samples were first preprocessed similar to the workflow used in IMP, i.e., using FastP v0.20.0[113]. Subsequently, the reads were deduplicated to avoid overlap and enhance computation efficiency using *clumpify.sh* from the BBmap suite v38.79[114]. Thereafter, any reads mapping to bacteria or viruses were removed by filtering the reads against a Kraken2 v2.0.9beta[115] maxikraken database available at https://lomanlab.github.io/mockcommunity/mc_databases.html. Only reads that were unknown or mapping to eukaryotes were retained and concatenated. This was followed by another round of deduplication using *clumpify.sh*. The concatenated reads were assembled using MEGAHIT v1.2.7 with the following options: *-kmin-1pass -m 0.9 -k-list 27,37,47,57,67,77,87 -min-contig-len 1000*. Following assembly, EukRep v0.6.7[116] was used for retrieving eukaryotic contigs with a minimum length of 2000 bp and the '*-m strict*' flag. These contigs were used for binning into MAGs as described herein.

Eukaryotic MAGs were binned using CONCOCT v1.1.0[117]. To do this, coverages were estimated for the contigs by mapping the reads of all samples against the contigs using the coverm v0.6.1 (https://github.com/wwood/CoverM) to generate bam files. These files were then used to generate a table with coverage depth information per sample. The protein coding genes of the MAGs was predicted with MetaEuk v4.a0f584d[118] with their in-house database made with MERC, MMETSP and Uniclust50 (http://wwwuser.gwdg.de/~compbiol/metaeuk/). The annotation was then subsequently done with eggNOG-mapper v2.1.0[119]. The completeness and contamination of the MAGs were assessed with Busco v5.0.0[120] and the eukaryotic lineage (255 genes). We determined their taxonomy by comparing the results of the EUKulele v1.0.3[107] and EukCC v0.3[121] along with homology comparisons with publicly available genomes not included in the previous tools by protein BLAST v2.10.0[122].

**Co-occurrence interaction networks**. Co-occurrence networks between the pro- and eukaryotic MAGs were constructed using an average of the distance matrices created from SparCC[123], Spearman's correlation and SpiecEasi[124], where the networks were constructed using the 'Meinshausen and Bühlmann (mb)' method. Nodes with fewer than two degrees were discarded to identify cliques with three or more interactions, while negative edges were removed to visualize only mutualistic relationships. The matrix was visualised using the igraph[125] R package. The largest component from the overall co-occurrence network was determined using the *components* module of the *igraph* package. Null model hypothesis was tested by assessing the distribution of the node degree and the respective probabilities of the occurrence network against those simulating the Erdos-Renyi, Barabasi-Albert, Stochastic-block null models[126]. The igraph package was also used for plotting the networks.

**Phylogenomics and pangenomes**. For the pangenome analyses, we collected all the bins taxonomically identified as *Polaromonas* spp. and used the pangenome workflow described by Meren *et al.* (http://merenlab.org/2016/11/08/pangenomics-v2) using anvi'o[111], along with NCBI[127] refseq genomes for comparison and an outgroup from the closely related *Rhodoferax* genus. The choice of *Polaromonas* spp. was based on its high abundance and prevalence within the epilithic biofilms. The accession IDs from the reference genomes obtained from NCBI are provided in the supplementary material. The pangenome was run using the *-min-bit 0.5, -mcl-inflation 10* and *-min-occurence 2* parameters, excluding the partial gene calls. A phylogenomic tree was built using MUSCLE v3.8.1551[128] and FastTree2 v2.1.10[129] on all single-copy gene clusters in the pangenome that were present in at least 30 genomes and had a functional homogeneity index below 0.9, and geometric homogeneity index above 0.9. The phylogenomic tree was used to order the genomes, the frequency of gene clusters (GC) to order the GC dendrogram. A phylogenomic bacterial tree of life containing the 47 high-quality MAGs along with 264 NCBI bacterial genomes was built based on a set of 74 single-copy genes using the GToTree v1.5.51[130] pipeline with the *-D* parameter, allowing to retrieve taxonomic information for the NCBI accessions. Briefly, HMMER3 v3.3.2[131] was used to retrieve the single-copy genes after gene-calling with Prodigal v2.6.3[132] and aligned using TrimAl v1.4.rev15[133]. The entire workflow is based on GNU Parallel v20210222[134].

**Data analyses and figures**. Figures for the study including visualizations derived from the taxonomic and functional components, were created using version 3.6 of the R statistical software package[135]. The maps indicating the collection sites were generated using the *ggmap*[136] package in R. *KEGGDecoder*[137] was used to assess enriched KEGG orthology (KO) IDs in comparison to 105 publicly available

metagenome sampled in various ecosystems at a global scale (Supplementary Data 3 and 6), which were processed using the IMP workflow. *DESeq2*[138] with FDR-adjustments for multiple testing were used to assess KOs significantly enriched in the GFS metagenomes compared to this comparison dataset. The volcano plot highlighting the significant KOs was generated using the *EnhancedVolcano*[139] R package. Figures from metabarcoding data were also generated in Rv3.6 using the *ggplot2*[140] package and were further annotated graphically using Inkscape[141] while the network plots were generated using the *igraph* v1.2.2 package.

**Reporting summary**. Further information on research design is available in the Nature Research Reporting Summary linked to this article.

## Data availability
Raw sequencing data samples and the MAGs are available at NCBI's sequence read archive under BioProject accession **PRJNA733707**. The Biosample accession IDs and the metadata associated with each sample are listed under Supplementary Data 6. A snippet of the results and source data generated and used in this study have been deposited in Zenodo at https://doi.org/10.5281/zenodo.5545722. Data used to generate the figures are also provided as a 'Source Data' file. Source data are provided with this paper.

## Code availability
The detailed code used for the downstream functional and growth analyses is available at https://git-r3lab.uni.lu/susheel.busi/nomis_pipeline and https://doi.org/10.5281/zenodo.6372573. The custom pipeline for eukaryote analyses can be found here: https://github.com/Mass23/NOMIS_ENSEMBLE/tree/coassembly. Subsequent binning and manual refinement of eukaryotic MAGs was done as described here: https://git-r3lab.uni.lu/susheel.busi/nomis_pipeline/-/blob/master/workflow/notes/MiscEUKMAGs.md.

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

## Acknowledgements

This research was funded by The NOMIS Foundation to TJB. SBB was supported by the Synergia grant (CRSII5_180241: Swiss National Science Foundation) to TJB. LdN and PW are supported by the Luxembourg National Research Fund (FNR; PRIDE17/11823097). RM and DD are supported by King Abdullah University of Science and Technology through baseline research funds to DD. We are thankful for the assistance of Audrey Frachet Bour, Lea Grandmougin, Janine Habier, Laura Lebrun (LCSB) and Emmy Marie Oppliger (EPFL) for laboratory support. We are grateful to Alex Washburne for his feedback on the draft, and we also acknowledge the valuable input from Rashi Halder at the LCSB Sequencing Platform with respect to library preparation. We are equally grateful for the valuable insights into metagenomic processing from Patrick May, Anna Heintz-Buschart, and Cedric Christian Laczny, and especially Valentina Galata with the python scripts and Snakemake workflows. The computational analyses presented in this paper were carried out using the HPC facilities at the University of Luxembourg (https://hpc.uni.lu)[142].

## Author contributions

SBB, MB, SF, HP, PW, and TJB conceived of the project. MiST, MT, VDS, MaSc, and HP conducted the fieldwork. PP, SF and SBB extracted DNA, while SBB and PP prepared the metagenomic and metabarcoding libraries, and RM and DD performed the sequencing. SBB conceptualized the data analyses, while SBB, MB, SF, GM and LE performed the analyses. LdN contributed to the python scripts and Snakemake workflows for the analyses. SBB, MB, TJK, PW and TJB wrote the manuscript with significant input and editing from all coauthors.

## Competing interests

The authors declare no competing interests.
