## [Peer Review File · Nature Communications]

REVIEWER COMMENTS

Reviewer #1 (Remarks to the Author):

The manuscript presents a detailed microbial community analysis of a unique habitat, glacial streams systems from two geographically distinct sites, New Zealand and the Caucasus mountains during the spring and autumn seasons. Communities were analysed by amplicon and metagenome sequencing, where the latter was used to infer functions of the communities. The data shows that the communities are distinct for both locations and that there is little similarity between communities from the riverbed/sand relative to biomass collected from boulders. One of the conclusions drawn throughout the manuscript is that these communities and their associated genes are reflective of how the microorganisms manage to exploit 'windows of opportunity' to dominate these habitats.

Conceptually this is very interesting. However, it is not clear what conclusions can be drawn relative to the windows of opportunity as there is not sufficient temporally resolved data to be sure that these communities and genes represent anything unique in terms of adaptation and the ability to exploit resources that is distinct from the opportunities a pathogen might have in colonising a host for example. It would seem that such conclusions would require an analysis that looks at founder effects vs genuine adaptive pathways specific for these habitats. If I follow the discussion points lines 221 to 266, the authors note a number of pathways that appear to be positively correlated between heterotrophs and algae, but when comparing the glacial stream community with MAGs from other ecosystems, the only pathways that are unique to the glacial streams are QS, B12 and thiamine biosynthesis, rather than those correlated to algal heterotroph interactions. This would appear to imply that adaptation to glacial streams is not vastly different to other habitats and their associated opportunities for colonisation.

The MAG and pathway analysis appears to be aggregated from all of the metacommunity data from both geographies and time points. Given that the communities seem to be so quite different between New Zealand and the Caucasus mountains, is it appropriate to do so given those communities would not be physically able to interact. It would be of interest to look at the site specific network analyses to understand what is unique or conserved in the two geographies.

Are there estimates of total biomass and if so, how do they compare with other oligotrophic vs copiotrophic environments? This question is based on the assertion that the communities overcome oligotrophic environments, which implies that there is more biomass than would be predicted.

It would be great if there were some physical/chemical measurements such as temperatures, oxygen and light levels, TOC/DOC or other nutrients including sulfur, and nitrogen species as well as extracts of the biomass, e.g. EPS, to help support the many hypotheses put forward based on the MAG analyses.

Line 98-99, something is missing, e.g. 'high' what?

Reviewer #2 (Remarks to the Author):

In the manuscript submitted by Busi et al., the researchers report on the microbial diversity found within biofilms that periodically form on different substrates in glacier fed-streams. Using standard SSU rRNA surveys (targeting either prokaryotes or eukaryotes), they compare microbial community structure between sediment biofilms and boulder biofilms in mountains of New Zealand and Russia. They then go

on to use metagenomics to further explore the structure and function of the boulder biofilms using genomes reconstructed from metagenomic datasets.

The central findings from the research are that there are a diversity of Cyanobacteria and eukaryotic algae in the biofilms that are serving as primary producers and that there are heterotrophic bacteria that interact with these phototrophs through degradation of organic compounds they produce. Somewhat unsurprisingly there is also evidence for proteorhodopsin-driven phototrophy. Other results include the identification of aquatic fungi, evidence for chemolithotrophy in the bacterial community, and a diversity of viruses mainly bacteriophage that would target the bacterial communities. Finally, they investigate how bacteria have adapted to the “extreme” environmental conditions of the glacier fed streams (GFS) using *Polaromonas* as a model. Using their own *Polaromonas* genomes and others that are publically available, they generate a pangenome and look for genes (or functional categories) that are associated with the GFS populations. Genes associated with cold stress, phage resistance, and perhaps UV radiation resistance were identified. An expansion of the analysis to the full genomes dataset found evidence for these genes being common across phylogenetic groups inhabiting the GFS.

Overall, the study is well done and the manuscript is clearly written. The findings highlighted in the paragraph above or more or less expected based on what we already know about aquatic microbiomes in general, but this is one of the first studies to report on the structure and function of these understudied GFS systems, so the work is generally novel.

Comments

1. I suggest the authors tone down on their language in places. I’m not sure the work is really “unprecedented” or the sequencing is “high resolution” This is pretty standard work for the field of metagenomics these days.
 2. I understand that the GFS biofilms are understudied and as a microbial ecologist I am deeply interested in the fundamental structure and function of biofilms, particularly in comparison to pelagic communities. But do these GFS biofilms serve any greater development of the ecosystem? Or is this microbial-dominated? Just wondering if climate change will only influence the microbial component of these ecosystems or might propagate up to changes in invertebrates?
 3. Line 147. Suggestion: you could compare to the GTDB and get an accurate estimate of how many new species, or higher taxonomic ranks are represented by the MAGs
 4. Lines 188-197. I don’t follow the logic here. So maybe think about rephrasing. There’s a negative relationship between predicted growth rate and median KEGG module completion? So, what about if whole metabolic modules are missing? For vitamin biosynthesis for example? Also how problematic is the incomplete nature of the MAGs. Completeness estimates are based on the core gene set, so you could be missing many other metabolic genes.
- Lines 231-234. This seems to go against the core idea that in this extreme environment photosynthetic production by Cyanobacteria is providing the limited amount of organic material for heterotrophs. Now Cyanobacteria are consuming it? Also, please add a reference for the “Widespread occurrence of mixotrophy in plankton communities....” sentence.

Reviewer #3 (Remarks to the Author):

The ms “Genomic and metabolic adaptations of biofilms to ecological windows of opportunities in glacier-fed streams” perfectly fits in the wake of modern ecological research on ecology of the cryosphere. In particular, the understanding of the role of ecological windows of opportunity represent a currently hot topic in cold water research.

The ms is based on a sound, state of the art methodological approach and, being based on the combination of complementary metabarcoding and metagenomic analyses, it provides a huge amount of valuable information. Some information is novel, while some results corroborates previous intuitions that were partially based on general ecological-principles (e.g. diversified metabolic pathways support diversity that in turn supports community resilience and resistance to environmental stressors). To this regard, one of the major contribution of the ms is to demonstrate that key adaptive traits of the GFSs microbiota are underpinned by genomic features.

In addition, the ms supports to the hypothesis of functional relationships between different microbial domains (e.g. heterotrophic bacteria and algae). Although these relationships have been guessed for a long time, a statistics-based demonstration represents a fundamental progress in environmental microbiology and ecology of glacial-fed running waters. Moreover the key ecological importance of cross-domain interactions is particularly stressed, along with the adaptive potential that epilithic microorganisms have developed to exploit the window of opportunity in GFSs.

In conclusion, I'm convinced the ms is worth publication, as it provides a wide and detailed picture on genomic underpinning of ecology and functionality of GFSs microbial communities.

I don't completely agree with the hypothesis that “epilithic biofilms may typify a ‘closed system’, where both carbon and nutrients are efficiently recycled”, since no natural ecosystem is completely closed in terms of matter fluxes, even less a running water ecosystem. However, I do agree with the hypothesis that the GFSs microbiota has the capacity to exploit opportunistically (i.e. during the short windows of opportunity) and very efficiently the extremely diluted resources thanks to enhanced in situ recycling capacity that are provided by diversified metabolic pathways, cross-domain interaction etc...

Although the paper aims to shed light on the genomic basis allowing epilithic biofilms to thrive during windows of opportunity in GFSs, I feel the authors should also provide at least some hypotheses on mechanisms possibly involved in the microbial survival outside WOS. This might set the path for further analyses and provide better balance and completeness to the huge result-set provided by the ms.

The figures are well done and all necessary, but it is difficult to keep track of the numerous plots (a, b, c,d...) in the different figures, also since Figs and Supp. Figs occurs close together in many text blocks (e.g. Fig. 3 being mentioned close to Suppl. 3). Mentions of Figs. is quite tricky in the paragraph “Genomic underpinnings of algae-bacteria metabolic interactions”, as there is insufficient match between text and figure content. I feel a simplification of the figure numbering and/or figure mentioning in the text as necessary to improve the reading and understanding of the ms.

Further comments/suggestions are listed here below.

L 35-36: I suggest to change the sentence in “The wide occurrence of rhodopsins, besides chlorophyll, across metagenome-assembled genomes (MAGs), highlights...

L 51: Why should be spring and autumn window of opportunity characterized by high nutrient availability? Please explain. And ... do the authors intend soluble nutrients? Particulate nutrients are often very abundant in glacier-fed stream, though hardly biologically available.

L53: Although subsidies of organic matter from the catchment are usually missing, it has been often demonstrated that glacial stream may be reach in highly available DOC of terrestrial origin (e.g. from ancient soils covered by the glacier). I see this point as relevant, as this organic source can support a “base-line” heterotrophic community all year long, although the window of opportunities are characterized by a dominant local primary productivity.

L77: ... allow biofilm to persist ...

L 89: ...which, similarly to the phycosphere, may...

L 109-112: I suggest moving the reasons for conducting the survey in MFS at the Earth antipodes to the introduction, as the question quickly arises to the reader. The sentence on the sampling time may be instead moved to the method section, while being only shortly reminded in the result section.

L 117: add a short explanation of NMDS and db-RDA adopted criteria in the method section.

L 121-124: the author should consider also habitat-related factors to explain the higher diversity of epipsammic communities. Although the epipsammic environment is physically unstable (due to the water flow), it may provide higher availability of organic matter (sand and silt are typically found in sheltered reaches with low flow velocity that favours particle sedimentation). This may make the habitat less oligotrophic and less homogeneous (respect to epilithon) and promote diversified metabolic paths and, consequently, biodiversity.

L150: I suggest to avoid the colour gradient for the taxon abundance as, at a first glance, red and blue may be confused with the colours assigned to the two study districts.

L 128: I suggest having a look also at other two recent papers on glacial biodiversity in N-America (Fegel et al., 2016) and N-Alps (Tolotti et al., 2020), as both stress higher bacterial biodiversity in surface-sediments than in epilithon of glacier-fed streams.

L 143: I don't find very appropriate the reference to medium-to-high quality of metagenome-assembled genomes (MAGs) in the text, while Fig. 2 is restricted to high quality MAGs, as it may generate confusion. I suggest either to mention both quality levels in the text, or to justify the difference in the legend of Fig. 2.

L207: It is quite difficult to identify the different taxa plotted in Fig. 2 due to the high number of taxa and the colour palette used. I suggest using also different symbols for the major bacterial groups (e.g. Bacteroidota, Proteobacteria) and Eukaryota.

L224: add a citation to this sentence, although it may sound trivial, just to benefit the more generic reader.

L229: suggested reformulation: However, based on the presence of the EEA genes also in phototrophic

genera, especially among Cyanobacteria, we cannot discount the possibility of mixotrophy in the epilithic biofilms (Supp. Fig. 4a), also in charge to other abundant members of the epilithic microbiome (Supp. Fig. 1c-d).

I don't see the necessity to refer here to Fig. 1c-d, without mentioning the prokaryotic (other than cyanos) and eukaryotic groups that may possibly perform mixotrophy. The sentence needs a further reformulation, since it is well known that Ochrophyta, Dinflagellata Cryptophyta (and likely others) living in high altitude, ultra-oligotrophic lacustrine ecosystems are typically mixotrophic. The novelties to be clearly stressed here are: 1) also cyanobacteria can perform mixotrophy, 2) mixotrophy is widespread also in running water due to algal groups that resulted rather abundant in the epilithic microbiota, and to cyanos. Algal mixotrophy has been demonstrated mainly in alpine/sub-polar/polar lake plankton, and a couple of citations should be added here, e.g. these classical papers by Rhode et al., 1966; Porter, 1988; Gervais, 1997; Jsaksson, 1998, and more recent ones.

L230: I guess Supp. Fig. 4b was meant here.

L240: add reference to Fig. 4c, that at present mentioned later.

L 244: Fig. 4c is mentioned before 4a and 4b (first mention around line 260). Reorganization of the plots within Fig. 4 seems necessary, but I fear the authors wanted to mention a different Fig. here. Possibly Suppl. Fig 1d?

L 288: ...that sulfates derived from sulfide oxidation...

L 298.299: ...and, to a lesser extent, denitrification, as major pathways (Fig. 4d).

L305: Fig. 4a was intended here?

L 338: ... cryophilic bacteria (such as *Janthinobacterium* spp.) develop...

L 344: ... the maintenance of the cell membrane in a liquid-crystalline state...

L 365: genomic adaptation to harsh GFS habitat should be also included in this conclusion as the third pillar allowing biofilm to thrive during (and likely outside) windows of opportunity in GFSs.

L 357 and 817: authors' names are missing in the citation 66

L 354-56: what is "an overall higher copy number of genes involved in counteracting osmotic and oxidative stress?" Did the authors test the significance of this higher proportion? As other, though smaller, bacterial groups show high copy numbers of these genes, I think a mention would be worth, although there may be no supporting literature to this observation.

L 428: a citation necessary here.

L 491: ... SpiecEasi106, where ...

L 528: add any reference to the free software Inkscape (or at least provide a link).

Response letter to reviewers' comments

The reviewers' comments are highlighted within boxes and the authors' responses to the reviewers' comments are listed in italic blue text below. Additionally, the authors numbered the comments (e.g., 1.1 for comment no. 1 from reviewer 1) to improve readability. We have also highlighted the line numbers where the appropriate changes have been made.

Reviewer #1 (Remarks to the Author):

The manuscript presents a detailed microbial community analysis of a unique habitat, glacial streams systems from two geographically distinct sites, New Zealand and the Caucasus mountains during the spring and autumn seasons. Communities were analysed by amplicon and metagenome sequencing, where the latter was used to infer functions of the communities. The data shows that the communities are distinct for both locations and that there is little similarity between communities from the riverbed/sand relative to biomass collected from boulders. One of the conclusions drawn throughout the manuscript is that these communities and their associated genes are reflective of how the microorganisms manage to exploit 'windows of opportunity' to dominate these habitats.

Response: We thank the reviewer for the recognition of our work and the useful comments allowing us to improve our study. In light of the major concerns expressed by the reviewer, please find our point-by-point responses below.

Comment 1.1. Conceptually this is very interesting. However, it is not clear what conclusions can be drawn relative to the windows of opportunity as there is not sufficient temporally resolved data to be sure that these communities and genes represent anything unique in terms of adaptation and the ability to exploit resources that is distinct from the opportunities a pathogen might have in colonising a host for example. It would seem that such conclusions would require an analysis that looks at founder effects vs genuine adaptive pathways specific for these habitats. If I follow the discussion points lines 221 to 266, the authors note a number of pathways that appear to be positively correlated between heterotrophs and algae, but when comparing the glacial stream community with MAGs from other ecosystems, the only pathways that are unique to the glacial streams are QS, B12 and thiamine biosynthesis, rather than those correlated to algal heterotroph interactions. This would appear to imply that adaptation to glacial streams is not vastly different to other habitats and their associated opportunities for colonisation.

Response: We thank the reviewer for this comment. We also agree with the reviewer that more temporally resolved data would be helpful to nail down some of our observations related to the windows of opportunity (WOPs). This would have been beyond the scope of our study as it would have involved sampling at multiple timepoints (under often difficult conditions and at logistically difficult sites to access) in the New Zealand and Caucasus streams. It was our aim to sample biofilms in the GFSs during the vernal (New Zealand) and autumnal (Caucasus) WOPs. We have now clarified this in the revision in lines 682-690. We have also clarified that the biofilms growing on the boulders (i.e., epilithic) may well extend beyond the typical WOP — at least until high discharge and/or snow cover.

As highlighted by the reviewer, several pathways associated with photo-heterotrophic interactions were found in the MAGs obtained from the epilithic biofilms. On the other hand,

the comparisons with other ecosystems were performed at the whole 'metagenome' levels. We have clarified this disparity in the text in line 459.

It must also be noted that thiamine biosynthesis is relevant for photo-heterotrophic interactions, as highlighted by Zheng *et al.* (doi: 10.1128/mBio.03261-19). In this context, most algae are auxotrophs for vitamin B12 which are potentially provided by heterotrophic bacteria (doi: 10.1111/j.1462-2920.2012.02733.x). Similarly, quorum sensing mechanisms allow for social interactions within biofilms as reviewed by Li *et al.* (doi: 10.3390/s120302519). The reviewer is correct in suggesting that there are few pathways "unique" to the glacier-fed streams. However, it must also be noted, as highlighted in Fig. 3d, that several pathways are significantly enriched/abundant compared other ecosystems, highlighting the extensive adaptations required by microorganisms in glacier-fed streams (GFSs).

Comment 1.2. The MAG and pathway analysis appears to be aggregated from all of the metacommunity data from both geographies and time points. Given that the communities seem to be so quite different between New Zealand and the Caucasus mountains, is it appropriate to do so given those communities would not be physically able to interact. It would be of interest to look at the site specific network analyses to understand what is unique or conserved in the two geographies.

Response: We thank the reviewer for this crucial comment, which we have now addressed by including additional network analyses for the Southern Alps and Caucasus separately. The overall network topologies were similar to the aggregated and the independent networks, with similar taxa (both pro- and eukaryotes) forming the nodes of the largest connected component. This is now reflected in the updated Supplementary Figure 3 and revised manuscript in lines 400-404.

Comment 1.3. Are there estimates of total biomass and if so, how do they compare with other oligotrophic vs copiotrophic environments? This question is based on the assertion that the communities overcome oligotrophic environments, which implies that there is more biomass than would be predicted.

Response: We are grateful for this comment. While we have bacterial cell counts, EPS, and chlorophyll α for the epipsammic biofilms, we do not have these data for the epilithic biofilms. This is because the latter would have to be normalized by surface area of the boulder, the former by mass. So, they would be hardly comparable even if we would normalize the epipsammic biomass by surface area under various assumptions. Additionally, epilithic biofilms are highly heterogenous within and among the streams due to the unequal presence of boulders in each GFS. As a result, we chose to sample epilithic biofilms opportunistically to ensure sample collection, albeit with the trade off with respect to our inability to estimate the biomass quantitatively.

Comment 1.4. It would be great if there were some physical/chemical measurements such as temperatures, oxygen and light levels, TOC/DOC or other nutrients including sulfur, and nitrogen species as well as extracts of the biomass, e.g. EPS, to help support the many hypotheses put forward based on the MAG analyses.

Response: We agree with the reviewer's assessment, and we have now updated the information on the GFS water chemistry in the Supplementary Information (Supplementary

Table 5). We prefer to abstain from showing biomass estimates for the epipsammic biofilms, given that we do not have corresponding estimates for the epilithic biofilms (please, refer to previous comment).

Comment 1.5. Line 98-99, something is missing, e.g. 'high' what?

Response: This has been corrected.

Reviewer #2 (Remarks to the Author):

In the manuscript submitted by Busi et al., the researchers report on the microbial diversity found within biofilms that periodically form on different substrates in glacier fed-streams. Using standard SSU rRNA surveys (targeting either prokaryotes or eukaryotes), they compare microbial community structure between sediment biofilms and boulder biofilms in mountains of New Zealand and Russia. They then go on to use metagenomics to further explore the structure and function of the boulder biofilms using genomes reconstructed from metagenomic datasets.

The central findings from the research are that there are a diversity of Cyanobacteria and eukaryotic algae in the biofilms that are serving as primary producers and that there are heterotrophic bacteria that interact with these phototrophs through degradation of organic compounds they produce. Somewhat unsurprisingly there is also evidence for proteorhodopsin-driven phototrophy. Other results include the identification of aquatic fungi, evidence for chemolithotrophy in the bacterial community, and a diversity of viruses mainly bacteriophage that would target the bacterial communities. Finally, they investigate how bacteria have adapted to the “extreme” environmental conditions of the glacier fed streams (GFS) using *Polaromonas* as a model. Using their own *Polaromonas* genomes and others that are publically available, they generate a pangenome and look for genes (or functional categories) that are associated with the GFS populations. Genes associated with cold stress, phage resistance, and perhaps UV radiation resistance were identified. An expansion of the analysis to the full genomes dataset found evidence for these genes being common across phylogenetic groups inhabiting the GFS.

Overall, the study is well done and the manuscript is clearly written. The findings highlighted in the paragraph above are more or less expected based on what we already know about aquatic microbiomes in general, but this is one of the first studies to report on the structure and function of these understudied GFS systems, so the work is generally novel.

Response: We thank the reviewer for the encouraging assessment, including the recognition of the in-depth analyses. We are also grateful for the critical and insightful comments. To address these comments, please find below our point-by-point responses.

Comment 2.1. I suggest the authors tone down on their language in places. I'm not sure the work is really “unprecedented” or the sequencing is “high resolution” This is pretty standard work for the field of metagenomics these days.

Response: We thank the reviewer for this comment and we have adjusted the language as suggested.

Comment 2.2. I understand that the GFS biofilms are understudied and as a microbial ecologist I am deeply interested in the fundamental structure and function of biofilms, particularly in comparison to pelagic communities. But do these GFS biofilms serve any greater development of the ecosystem? Or is this microbial-dominated? Just wondering if climate change will only influence the microbial component of these ecosystems or might propagate up to changes in invertebrates?

Response: We are much grateful for this comment. The reviewer is correct in stating that the GFSs are dominated by microbes — they form the basis of a ‘moderate’ food web and do sustain invertebrates. Milner *et al.* (doi: 10.1073/pnas.1619807114) have speculated that GFS ecosystems may become more autotrophic beyond peak flow as glacier shrink (doi: 10.1038/s41559-019-1042-8). We relate to this in our manuscript, also highlighting clearly that the epilithic biofilms form the foundation of the ‘green’ food web in the GFSs. Thereby, biofilms regulate critical ecosystem processes, and there is first evidence that they could stabilize sandy sediments — perhaps not in the main GFS channels, but certainly in the tributaries within the proglacial floodplains. We recognize that this was not sufficiently clear in the original submission and have improved this part of the introduction.

Comment 2.3. Line 147. Suggestion: you could compare to the GTDB and get an accurate estimate of how many new species, or higher taxonomic ranks are represented by the MAGs

Response: We thank the reviewer for this suggestion. As highlighted in the Methods in line 786, the taxonomic affiliation of the MAGs was determined by using the GTDBtk methodology. Subsequently, 15 out of the 49 high-quality MAGs (>90% completion and <5% contamination) were only identified up to the family level. On the other hand, only genus level taxonomic affiliations were retrieved for the remaining 34 MAGs. It is likely that these are novel species or genera that have not been previously identified or studied. This information has now been included in the revised manuscript in lines 256-274 and 982.

Comment 2.4. Lines 188-197. I don’t follow the logic here. So maybe think about rephrasing. There’s a negative relationship between predicted growth rate and median KEGG module completion? So, what about if whole metabolic modules are missing? For vitamin biosynthesis for example? Also how problematic is the incomplete nature of the MAGs. Completeness estimates are based on the core gene set, so you could be missing many other metabolic genes.

Response: We thank the reviewer for the suggestion and we have reformulated the text accordingly. We have addressed the potential caveat with incomplete MAGs by accounting for completion and contamination in the regression analyses. This information along with the updated statistics have now been included in the revised manuscript in lines 331-335.

Comment 2.5. Lines 231-234. This seems to go against the core idea that in this extreme environment photosynthetic production by Cyanobacteria is providing the limited amount of organic material for heterotrophs. Now Cyanobacteria are consuming it? Also, please add a reference for the “Widespread occurrence of mixotrophy in plankton communities....” sentence.

Response: We thank the reviewer for this interesting observation. Cyanobacteria have indeed been shown to demonstrate mixotrophic functions in marine (doi: 10.1111/1462-2920.14111) and polar regions (doi: 10.3389/fmars.2018.00273). And as depicted in Supplementary Figure 4b, some of our Cyanobacterial MAGs indeed encode genes associated with mixotrophy. It is, therefore, plausible that some Cyanobacteria consume some of the organic material synthesized within the biofilms. We leave this as a speculation rather than as a fact for which metatranscriptomics/metaproteomics may be required for definitive confirmation. However, we do not believe that the possibility of mixotrophy would run against the core idea of a metabolic link as suggested by the reviewer. Rather, we think that the diversification of various metabolic strategies, including their redundancy across several MAGs including Cyanobacteria, would benefit the various biota to ‘build’ a resilient biofilm in an extreme environment, especially during and outside of the windows of opportunity — please, see also the general comment from reviewer #3. This information has now been elaborated upon in the revised manuscript in lines 433-437. Furthermore, as suggested by the reviewer, a citation has been added in lines 438-440.

Reviewer #3 (Remarks to the Author):

The ms “Genomic and metabolic adaptations of biofilms to ecological windows of opportunities in glacier-fed streams” perfectly fits in the wake of modern ecological research on ecology of the cryosphere. In particular, the understanding of the role of ecological windows of opportunity represent a currently hot topic in cold water research.

The ms is based on a sound, state of the art methodological approach and, being based on the combination of complementary metabarcoding and metagenomic analyses, it provides a huge amount of valuable information. Some information is novel, while some results corroborates previous intuitions that where partially based on general ecological-principles (e.g. diversified metabolic pathways support diversity that in turn support s community resilience and resistance to environmental stressors). To this regard, one of the major contribution of the ms is to demonstrate that key adaptive traits of the GFSs microbiota are underpinned by genomic features.

In addition, the ms supports to the hypothesis of functional relationships between different microbial domains (e.g. heterotrophic bacteria and algae). Although these relationships have been guessed for a long time, a statistics-based demonstration represents a fundamental progress in environmental microbiology and ecology of glacial-fed running waters. Moreover the key ecological importance of cross-domain interactions is particularly stressed, along with the adaptive potential that epilithic microorganisms have developed to exploit the window of opportunity in GFSs.

In conclusion, I’m convinced the ms is worth publication, as it provides a wide and detailed picture on genomic underpinning of ecology and functionality of GFSs microbial communities.

Response: We thank the reviewer for highlighting the major findings and appreciate their constructive feedback with respect to improving the manuscript. In light of the reviewer’s

suggestions, we have highlighted below, point-by-point, a proposal for revisions and additions that address the feedback.

Comment 3.1. I don't completely agree with the hypothesis that "epilithic biofilms may typify a 'closed system', where both carbon and nutrients are efficiently recycled", since no natural ecosystem is completely closed in terms of matter fluxes, even less a running water ecosystem. However, I do agree with the hypothesis that the GFSs microbiota has the capacity to exploit opportunistically (i.e. during the short windows of opportunity) and very efficiently the extremely diluted resources thanks to enhanced in situ recycling capacity that are provided by diversified metabolic pathways, cross-domain interaction etc...

Response: We are very grateful for this critical comment. Without any doubt, we do agree with this reviewer that there is no 'closed system' in nature, certainly not in streams – all are a matter of scales. In streams, benthic biofilms enhance transient storage of water and contained solutes, which facilitates the internal recycling of matter (doi: 10.1038/nature02152 and 10.1128/AEM.69.9.5443-5452.2003). This internal recycling, aided by various metabolic interactions, is what we intended to highlight. We have changed the respective text in lines 622-623 accordingly to clarify this.

Comment 3.2. Although the paper aims to shed light on the genomic basis allowing epilithic biofilms to thrive during windows of opportunity in GFSs, I feel the authors should also provide at least some hypotheses on mechanisms possibly involved in the microbial survival outside WOS. This might set the path for further analyses and provide better balance and completeness to the huge result-set provided by the ms.

Response: We acknowledge the reviewer's valuable comment and in accordance with their suggestion expanded the section in lines 638-665. The revised manuscript describes the potential pathways and adaptation strategies that may help the epilithic biofilms alleviate the harsh conditions outside of the windows of opportunity. Specifically, mixotrophy as observed within Cyanobacteria would be advantageous during periods outside of the windows of opportunity. Along with these, we highlight our observations with respect to cold adaptation genes involved in cell membrane alterations and lipid composition within the MAGs. This is in line with a report by Tribelli and López (doi: 10.3390/life8010008). Simultaneously, we elucidate the potential role which genes associated with counteracting oxidative stress may play, given the cryophilic temperatures outside of the windows of opportunity. The overall diversity of cold adaptation genes (76 in total) allows for generating key hypotheses that may be tested both experimentally and *in silico*, going forward.

Comment 3.3. The figures are well done and all necessary, but it is difficult to keep track of the numerous plots (a, b, c,d...) in the different figures, also since Figs and Supp. Figs occurs close together in many text blocks (e.g. Fig. 3 being mentioned close to Suppl. 3). Mentions of Figs. is quite tricky in the paragraph "Genomic underpinnings of algae-bacteria metabolic interactions", as there is insufficient match between text and figure content. I feel a simplification of the figure numbering and/or figure mentioning in the text as necessary to improve the reading and understanding of the ms.

Response: Based on the reviewer's suggestions, the appropriate figure numbers have been updated to match the described text.

Further comments/suggestions are listed here below.

Comment 3.4. L 35-36: I suggest to change the sentence in “The wide occurrence of rhodopsins, besides chlorophyll, across metagenome-assembled genomes (MAGs), highlights...”

Response: Added as suggested.

Comment 3.5. L 51: Why should be spring and autumn window of opportunity characterized by high nutrient availability? Please explain. And ... do the authors intend soluble nutrients? Particulate nutrients are often very abundant in glacier-fed stream, though hardly biologically available.

Response: We might expect higher dissolved N concentrations in spring due to the accumulation and subsequent concentration of N on the surface of the snowpack over the winter. Thus, as the melt season commences, the ‘first flushing’ of the system tends to have higher N concentrations than the rest of the year. In the case of P, much of the soluble fraction tends to bind to the fine particles suspended in the glacier meltwater (which as the review mentions is often not bioavailable), and a greater proportion of P may be locked in this particulate fraction as turbidity increases with greater glacier meltwater generation during summer. Soluble nutrients in general are diluted by meltwater as the melt season progresses, but concentrations may again rise somewhat as discharge declines at the end of the melt season in autumn. These are of course generalizations and is likely not the case for all glaciers streams everywhere. We have now expanded upon these ideas in the revised text on lines 55-62.

Comment 3.6. L53: Although subsidies of organic matter from the catchment are usually missing, it has been often demonstrated that glacial stream may be reach in highly available DOC of terrestrial origin (e.g. from ancient soils covered by the glacier). I see this point as relevant, as this organic source can support a “base-line” heterotrophic community all year long, although the window of opportunities are characterized by a dominant local primary productivity.

Response: This is an interesting comment indeed for which we are grateful. Indeed, previous work by the PIs lab and others (doi: 10.1038/ngeo618 and 10.1029/2019GL083424d) has shown mountain glaciers as a potential source of organic carbon to the downstream microbes, an observation that Fellman and colleagues (2015, *Limnology & Oceanography*) were able to expand to the GFS food web. However, we need to emphasize that the DOC concentration in GFSs is notoriously low (see Supplementary table 5). Therefore, while we tend to be cautious to make the point of a ‘base line’ resource for heterotrophs, we still believe that it may be an interesting notion.

Comment 3.7. L77: ... allow biofilm to persist ...

Response: Added as suggested.

Comment 3.8. L 89: ...which, similarly to the phycosphere, may...

Response: Corrected as suggested.

Comment 3.9. L 109-112: I suggest moving the reasons for conducting the survey in MFS at the Earth antipodes to the introduction, as the question quickly arises to the reader. The sentence on the sampling time may be instead moved to the method section, while being only shortly reminded in the result section.

Response: We thank the reviewer for the comment and have moved the lines as suggested. These lines can be found in the revised manuscript in lines 181-182 and 192-197.

Comment 3.10. L 117: add a short explanation of NMDS and db-RDA adopted criteria in the method section.

Response: As per the reviewer's suggestions, we have updated the Methods section and added details on data transformation and algorithm in the revised manuscript in lines 750-752.

Comment 3.11. L 121-124: the author should consider also habitat-related factors to explain the higher diversity of epipsammic communities. Although the epipsammic environment is physically unstable (due to the water flow), it may provide higher availability of organic matter (sand and silt are typically found in sheltered reaches with low flow velocity that favours particle sedimentation). This may make the habitat less oligotrophic and less homogeneous (respect to epilithon) and promote diversified metabolic paths and, consequently, biodiversity.

Response: This is a very good point for which we are grateful. Discharge in GFSs can vary tremendously from day to night, which induces cycles of transportation and sedimentation of sandy sediments. It is indeed this continuous source-sink mixing that may contribute to the higher diversity observed with the epipsammic biofilms. We have now better clarified these thoughts in the revision in lines 60-62.

Comment 3.12. L150: I suggest to avoid the colour gradient for the taxon abundance as, at a first glance, red and blue may be confused with the colours assigned to the two study districts.

Response: The figure has been updated as suggested.

Comment 3.13. L 128: I suggest having a look also at other two recent papers on glacial biodiversity in N-America (Fegel et al., 2016) and N-Alps (Tolotti et al., 2020), as both stress higher bacterial biodiversity in surface-sediments than in epilithon of glacier-fed streams.

Response: We thank the reviewer for these suggestions and have discussed their findings in light of our observations in lines 230-233 and 236-238 in the revised manuscript.

Comment 3.14. L 143: I don't find very appropriate the reference to medium-to-high quality of metagenome-assembled genomes (MAGs) in the text, while Fig. 2 is restricted to high quality MAGs, as it may generate confusion. I suggest either to mention both quality levels in the text, or to justify the difference in the legend of Fig. 2.

Response: We thank the reviewer for this comment and have clarified the use of medium- and high-quality MAGs in the text in lines 256-269 and in the legend for figure 2 in line 988-989.

Comment 3.15. L207: It is quite difficult to identify the different taxa plotted in Fig. 2 due to the high number of taxa and the colour palette used. I suggest using also different symbols for the major bacterial groups (e.g. Bacteroidota, Proteobacteria) and Eukaryota.

Response: Figure 2 has been updated to make clear distinctions between the most abundant Phyla, including all other taxa found in the epilithic biofilm metagenomes.

Comment 3.16. L224: add a citation to this sentence, although it may sound trivial, just to benefit the more generic reader.

Response: Citation added as suggested.

Comment 3.17. L229: suggested reformulation: However, based on the presence of the EEA genes also in phototrophic genera, especially among Cyanobacteria, we cannot discount the possibility of mixotrophy in the epilithic biofilms (Supp. Fig. 4a), also in charge to other abundant members of the epilithic microbiome (Supp. Fig. 1c-d).

Response: Reformulated as suggested.

Comment 3.18. I don't see the necessity to refer here to Fig. 1c-d, without mentioning the prokaryotic (other than cyanos) and eukaryotic groups that may possibly perform mixotrophy. The sentence needs a further reformulation, since it is well known that Ochrophyta, Dinflagellata Cryptophyta (and likely others) living in high altitude, ultra-oligotrophic lacustrine ecosystems are typically mixotrophic. The novelties to be clearly stressed here are: 1) also cyanobacteria can perform mixotrophy, 2) mixotrophy is widespread also in running water due to algal groups that resulted rather abundant in the epilithic microbiota, and to cyanos. Algal mixotrophy has been demonstrated mainly in alpine/sub-polar/polar lake plankton, and a couple of citations should be added here, e.g. these classical papers by Rhode et al., 1966; Porter, 1988; Gervais, 1997; Jsaksson, 1998, and more recent ones.

Response: We thank the reviewer for this valuable comment and have revised the manuscript with the suggested literature. We have also elaborated on the mixotrophic role of cyanobacteria within the epilithic biofilms in GFSs in lines 433-441 and 653-654.

Comment 3.19. L230: I guess Supp. Fig. 4b was meant here.

Response: The reviewer is correct that the figure reference was meant to be Supp. Fig. 4b and has been modified accordingly.

Comment 3.20. L240: add reference to Fig. 4c, that at present mentioned later.

Response: This has been clarified to indicate the correct figures, Fig. 3c and Supp. Fig. 4d.

Comment 3.21. L 244: Fig. 4c is mentioned before 4a and 4b (first mention around line 260). Reorganization of the plots within Fig. 4 seems necessary, but I fear the authors waned to mention a different Fig. here. Possibly Suppl. Fig 1d?

Response: In line with the response to Comment 3.20, this has been clarified.

Comment 3.22. L 288: ...that sulfates derived from sulfide oxidation...

Response: Corrected as suggested.

Comment 3.23. L 298.299: ...and, to a lesser extent, denitrification, as major pathways (Fig. 4d).

Response: Corrected as suggested.

Comment 3.24. L305: Fig. 4a was intended here?

Response: Figure references updated as suggested.

Comment 3.25. L 338: ... cryophilic bacteria (such as *Janthinobacterium* spp.) develop...

Response: Added as suggested.

Comment 3.26. L 344: ... the maintenance of the cell membrane in a liquid-crystalline state...

Response: Added as suggested.

Comment 3.27. L 365: genomic adaptation to harsh GFS habitat should be also included in this conclusion as the third pillar allowing biofilm to thrive during (and likely outside) windows of opportunity in GFSs.

Response: The conclusion has been reformulated as suggested in the revised manuscript in lines 666-669.

Comment 3.28. L 357 and 817: authors' names are missing in the citation 66

Response: The citation has been added as suggested.

Comment 3.29. L 354-56: what is “an overall higher copy number of genes involved in counteracting osmotic and oxidative stress?” Did the authors test the significance of this higher proportion? As other, though smaller, bacterial groups show high copy numbers of these genes, I think a mention would be worth, although there may be no supporting literature to this observation.

Response: We thank the reviewer for this comment and have updated the text to include the statistical analysis in lines 623-630. We have additionally included the per phylum statistical analyses with respect to mean copy numbers per genome per phyla as Supplementary table 7 in the revised manuscript.

Comment 3.30. L 428: a citation necessary here.

Response: Citation added as suggested

Comment 3.31. L 491: ... SpiecEasi106, where ...

Response: Added as suggested.

Comment 3.32. L 528: add any reference to the free software Inkscape (or at least provide a link).

Response: Added as suggested.

REVIEWERS' COMMENTS

Reviewer #1 (Remarks to the Author):

The authors have done a nice job in addressing the comments in the previous version and I have no further comments to add. Very nice study and quite interesting.

Reviewer #2 (Remarks to the Author):

Thank you for considering and addressing the comments previously provided.

Reviewer #3 (Remarks to the Author):

The Author's addressed all the comments by the three reviewers in an accurate way. Regarding my comments (rev#3) I'm completely satisfied with the authors' responses and rebuttal. I'm convinced that the ms has substantially improved, especially thanks to a more holistic interpretation of ecological and functional traits of glacial-fed-streams.

I have only a couple of smaller comments to the revised ms version.

L 43: ...imposed by the pronounced oligotrophy...

L56-62: the explanation of reasons behind the high nutrient concentrations in spring is ok, and it is useful considering that the readers may not be necessarily confident with the chemical setting of glacial-fed headwaters.

L156-160. Since these lines somehow spoil the paper conclusions. I suggest to write them in form of ecological question to be addressed in the ms.

Response letter to reviewers' comments

The reviewers' comments are highlighted within boxes and the authors' responses to the reviewers' comments are listed in italic blue text below. Additionally, the authors numbered the comments (e.g., 1.1 for comment no. 1 from reviewer 1) to improve readability. We have also highlighted the line numbers where the appropriate changes have been made.

Reviewer #1 (Remarks to the Author):

The authors have done a nice job in addressing the comments in the previous version and I have no further comments to add. Very nice study and quite interesting.

Response: We thank the reviewer for the recognition of our work and their comments which helped us to improve our study.

Reviewer #2 (Remarks to the Author):

Thank you for considering and addressing the comments previously provided.

Response: We thank the reviewer and appreciate their constructive feedback, thereby improving the manuscript.

Reviewer #3 (Remarks to the Author):

The Author's addressed all the comments by the three reviewers in an accurate way. Regarding my comments (rev#3) I'm completely satisfied with the authors' responses and rebuttal. I'm convinced that the ms has substantially improved, especially thanks to a more holistic interpretation of ecological and functional traits of glacial-fed streams.

Response: We thank the reviewer for their constructive feedback and enabling a holistic view of the ecological and functional traits of the microbiome within the glacier-fed streams.

Minor comments/suggestions:

Comment 3.1. L 43: ...imposed by the pronounced oligotrophy...

Response: As suggested by the reviewer, the sentence has been modified.

Comment 3.2. L56-62: the explanation of reasons behind the high nutrient concentrations in spring is ok, and it is useful considering that the readers may not be necessarily confident with the chemical setting of glacial-fed headwaters.

Response: We acknowledge the reviewer's valuable comment and thank them for accepting our explanation.

Comment 3.3. L156-160. Since these lines somehow spoil the paper conclusions. I suggest to write them in form of ecological question to be addressed in the ms.

Response: Based on the reviewer's suggestions lines 156-160 have been modified.